# Privacy Amplification in
# Differentially Private Zeroth-Order Optimization with Hidden States

**Eli Chien** [1] [2]  **Wei-Ning Chen** [3]  **Pan Li** [4]

## Abstract

Zeroth-order optimization has emerged as a promising approach for fine-tuning large language models under differential privacy (DP) and memory constraints. While privacy amplification by iteration (PABI) provides convergent DP bounds for first-order methods, establishing similar guarantees for zeroth-order methods remains an open problem. First-order PABI analysis relies on the fact that gradients are perturbed with isotropic noise, allowing privacy bounds to be iteratively tracked via shifted Rényi divergence. In contrast, DP zeroth-order methods inject scalar noise along random update directions to maintain utility. This anisotropic update fails standard shifted divergence frameworks, as the global Lipschitz property no longer holds almost surely. We provide the first convergent hidden-state DP bound for zeroth-order optimization by proposing a hybrid noise mechanism and a novel coupling analysis. We bypass the purely shifted-divergence approach by constructing a coupled auxiliary process, which circumvents the global Lipschitz barrier and yields a convergent privacy bound. Furthermore, our results induce better DP zeroth-order algorithmic designs that are previously unknown to the literature.

## 1. Introduction

The scaling laws (Kaplan et al., 2020) suggest that increasing the amount of training data and model size leads to corresponding increases in model capability. It has demonstrated success in many large-scale vision and language foundational models, which typically consist of billions to hundreds of billions of parameters. However, as model size grows, preserving a model's privacy (in particular, differential privacy (Dwork et al., 2006)) during the training phase becomes increasingly challenging. This challenge primarily stems from the significant increase in memory usage when optimizing the model via differentially private optimizers such as DP-SGD (Abadi et al., 2016). To DP-fy these first-order methods, the standard approach is to clip the gradients of each sample and then inject calibrated Gaussian noise in each training round. The per-sample gradient clipping step, however, incurs huge memory costs since the optimizer has to cache all the intermediate gradients of each sample during back-propagation[1].

To address this issue, recent works by Zhang et al. (2024a); Tang et al. (2024) consider differentially private zeroth-order optimization, which relies only on the evaluated loss (i.e., forward pass of the model) and does not require the computation of gradients. While there are several variants of zeroth-order optimization, they generally involve the following steps: in each training round, a random perturbation is generated on the model weights (e.g., sampling from an isotropic Gaussian distribution), and the step size is computed based on the loss values evaluated on the perturbed weights. To ensure DP, one could clip and add noise to the loss values associated with each sample. Since these zeroth-order optimization methods do not require the computation of gradient, they significantly improve both memory and computational costs. Indeed, Zhang et al. (2024a); Tang et al. (2024) successfully fine-tuned models with over 60B parameters with DP and the performance is comparable with the state-of-the-art DP first-order methods such as DP-LoRA (Yu et al., 2022).

Despite successfully scaling up DP training to models with tens of billions of parameters, the current privacy analysis of the zeroth-order optimization method in Zhang et al. (2024a); Tang et al. (2024) relies on the composition theorem, which accounts for the privacy budget spent during each training round. As a result, the privacy cost increases

---

[1]Department of Electrical Engineering, National Taiwan University, Taiwan [2]NTU Artificial Intelligence Center of Research Excellence (NTU AI-CoRE), Taiwan [3]Microsoft, USA [4]Department of Electrical and Computer Engineering, Georgia Institute of Technology, USA. Correspondence to: Eli Chien <elichien@ntu.edu.tw>.

*Proceedings of the $43^{rd}$ International Conference on Machine Learning*, Seoul, South Korea. PMLR 306, 2026. Copyright 2026 by the author(s).

---

[1]While some recent works have proposed methods to alleviate such costs (e.g., via ghost clipping (Li et al., 2022)), the increase in memory usage remains significant compared to non-DP training.

*linearly* with the number of steps, and hence a carefully chosen stopping point is required to balance privacy and utility. Note that such framework assumes that intermediate optimization steps are public, potentially leading to an overestimation of the true privacy cost when only the *last iterate* (i.e., the final model weights) is released.

In this work, we demonstrate that by concealing the intermediate optimization steps and slightly tweaking the perturbation and model update steps, one can "amplify" the privacy guarantees provided by the composition theorem. This type of privacy analysis, known as "privacy amplification by iteration" (PABI), leverages the hidden optimization trajectory to achieve stronger privacy. While PABI has been extensively studied for first-order methods (Feldman et al., 2018; Altschuler & Talwar, 2022; 2023; Chien et al., 2024a; Chien & Li, 2025), existing analyses do not extend directly to zeroth-order optimization.

Applying PABI to zeroth-order methods introduces unique challenges, primarily due to the structure of the DP noise. In first-order methods like DP-SGD, gradients are perturbed with isotropic Gaussian noise each round, and privacy analysis relies on a key shifted-reduction lemma (Feldman et al., 2018) to control Rényi divergence iteratively. However, in zeroth-order optimization, noise is applied to the step size along a random direction $u$, resulting in "anisotropic" noise, i.e., a scalar Gaussian perturbation along $u$ rather than in all directions. As a result, the standard shifted-reduction lemmas are no longer applicable.

One could add isotropic Gaussian noise on top of zeroth-order updates and apply existing analysis tools, but prior work has shown that doing so severely worsens the privacy-utility trade-off (Zhang et al., 2024a). In zeroth-order methods, discrepancies between adjacent datasets only appear along the update direction $u$, making noise in the orthogonal subspace $u^\perp$ wasteful. Furthermore, such a random update direction $u$ leads to a worse global Lipschitz constant for the zeroth-order update compared to its first-order counterpart. Directly applying the shifted-divergence analysis gives an undesired privacy bound.

To address this, we design a hybrid noise mechanism that combines the advantages of directional and isotropic noise, achieving tighter privacy bounds. Specifically, our method employs (1) a random $K$-dimensional update each step to reduce variance, (2) Gaussian perturbation on the step size, and (3) a small isotropic Gaussian noise component. To address the Lipschitz constant issue, we first derive the high probability pointwise Lipschitz bound, which provides a much smaller Lipschitz constant. However, such a pointwise bound cannot be applied to the shifted-divergence analysis. We hence propose a novel coupling technique by constructing an auxiliary process $\widetilde{W}$ that is "in the middle" of two original training processes $W, W'$ on the adjacent

datasets. This construction allows us to decompose the privacy analysis into two manageable parts: a total variation (TV) distance bound on $W, \widetilde{W}$ that captures the failure probability of the coupling, and a Rényi divergence bound on $\widetilde{W}, W'$ derived from the shifted Gaussian mechanism.

Our analysis reveals several privacy advantages in zeroth-order method overlooked by prior work. First, we highlight the role of $K$, the dimension of the update at each iteration. While larger $K$ is known to improve utility by reducing the variance of gradient estimation, previous DP zeroth-order studies (Zhang et al., 2024a; Tang et al., 2024) avoid large $K$ due to the $\widetilde{O}(\sqrt{K})$ increase in privacy cost under standard composition. However, under hidden-state analysis, we show that privacy loss actually *decays* with $K$ under a fixed utility constraint. Standard composition fails to capture this benefit, treating the privacy loss as independent of $K$ (Zhang et al., 2024a). Furthermore, our analysis shows that when $K > 1$, choosing orthonormal update directions $\{u_{t,k}\}_{k=1,..,K}$ significantly improves the privacy bound compared to using i.i.d. random directions – a benefit not recognized in the existing literature. These findings not only provide tighter privacy guarantees but also offer practical guidelines for designing better differentially private zeroth-order optimization algorithms.

**Our contributions.** Our main contributions can be summarized as follows:

- *PABI for Zeroth-Order Methods*: We develop the first privacy amplification by iteration (PABI) analysis tailored for DP zeroth-order optimization, demonstrating a constant privacy cost that saturates after a fixed number of training steps. Interestingly, our analysis sidesteps the shifted-divergence analysis by directly constructing the auxiliary process that fits the zeroth-order update.

- *Hybrid Noise Mechanism*: We propose a new noise mechanism that combines directional and isotropic noise to enable tighter privacy bounds.

- *Role of Update Dimension $K$*: We show that, contrary to prior beliefs, the privacy loss decays with the update dimension $K$ under hidden-state analysis, and that using orthonormal directions further improves privacy.

Missing proofs, discussions on limitations, and broader impacts are deferred to the appendices.

## 2. Preliminaries and Problem formulation

Let $\mathcal{D} = \{x_i\}_{i=1}^n \in \mathcal{X}^n$ be a training dataset of $n$ data points from some domain $\mathcal{X}$. We say a dataset $\mathcal{D}'$ is adjacent to $\mathcal{D}$ with replacement (denoted by $\mathcal{D} \simeq \mathcal{D}'$) if $\mathcal{D}'$ can be obtained by replacing one data point $x_i$ from $\mathcal{D}$ with an

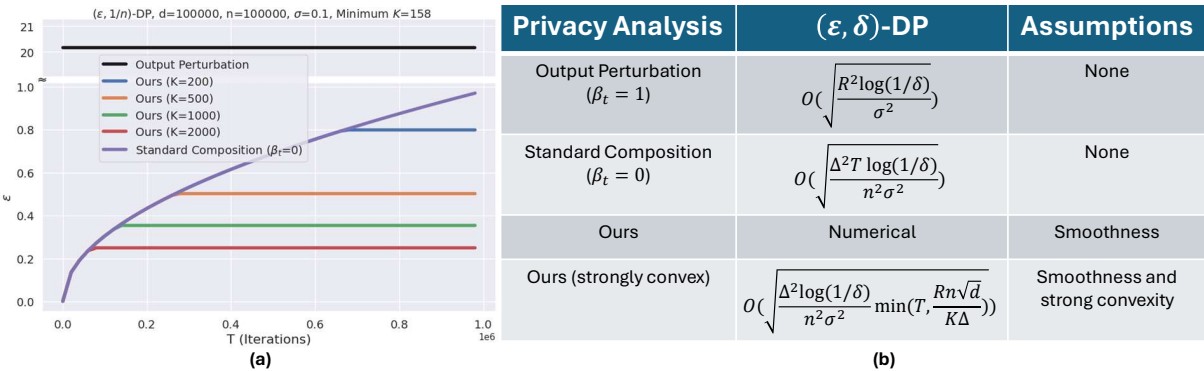

*Figure 1.* (a) The DP guarantees for smooth strongly convex losses over the bounded domain versus training iteration $T$, where the noise variance $\sigma^2$ is the same for all lines. $d$ is the number of parameters, and $n$ is the size of the training dataset. The output perturbation directly utilizes the Gaussian mechanism with sensitivity chosen to be the diameter of the bounded domain. The minimum $K$ indicates the lower bound on the choice of $K$ stated in Corollary 3.3. The non-convex case can be found at Appendix B. (b) Privacy bound $\varepsilon$ comparison to standard analysis for Noisy-ZOGD (4), where $\Delta$ is the clipped norm.

arbitrary point $x_i'$ in $\mathcal{X}$. One could also adopt a different adjacency definition as the one in Kairouz et al. (2021). We denote $[n] := \{1, 2, \cdots, n\}$ and $f\sharp\mu$ the pushforward of a distribution $\mu$ under a function $f$. We use $X_{i:j}$ as shorthand for the vector concatenating $X_i, \cdots, X_j$ and $X \overset{d}{=} Y$ to denote that random variables $X, Y$ are equal in distribution.

## 2.1. Differential Privacy

A randomized algorithm $\mathcal{A} : \mathcal{X}^n \to \mathcal{W}$ satisfies differential privacy (Dwork et al., 2006; Mironov, 2017), if the following holds:

**Definition 2.1** ((Rényi) Differential Privacy). A randomized algorithm $\mathcal{A} : \mathcal{X}^n \to \mathcal{W}$ is $(\varepsilon, \delta)$-differentially private (DP), if for any $\mathcal{D} \simeq \mathcal{D}'$ and any measurable $\mathcal{S} \subseteq \mathsf{range}(\mathcal{A})$,

$$\mathbb{P}\{\mathcal{A}(\mathcal{D}) \in \mathcal{S}\} \le e^\varepsilon \cdot \mathbb{P}\{\mathcal{A}(\mathcal{D}') \in \mathcal{S}\} + \delta.$$

$\mathcal{A}$ satisfies $(\varepsilon, \alpha)$-Rényi differential privacy (RDP) if $D_\alpha(\mathcal{A}(\mathcal{D})\|\mathcal{A}(\mathcal{D}')) \le \varepsilon$, where $D_\alpha$ denotes the Rényi divergence: $D_\alpha(\mu\|\nu) \triangleq \frac{1}{\alpha-1}\log\left(\int(\mu(x)/\nu(x))^\alpha \nu(x)dx\right)$ if $\mu \ll \nu$, and $\infty$ otherwise[2].

RDP enjoys several useful properties that will be utilized in our analysis.

**Lemma 2.2** (Post-processing property (Mironov, 2017)). *For any $\alpha > 1$, any function $f$, and any probability distribution $\mu, \nu$, $D_\alpha(h\sharp\mu\|h\sharp\nu) \le D_\alpha(\mu\|\nu)$.*

**Lemma 2.3** (Strong composition for Rényi divergence(Mironov, 2017)). *For any $\alpha > 1$ and any two sequences of random variables $X_1, \cdots, X_k$ and $Y_1, \cdots, Y_k$,*

it holds that

$$D_\alpha(X_{1:k}\|Y_{1:k}) \le$$
$$\sum_{i=1}^{k} \sup_{x_{1:i-1}} D_\alpha(X_i|_{X_{1:i-1}=x_{1:i-1}}\|Y_i|_{Y_{1:i-1}=x_{1:i-1}}). \quad (1)$$

We will also need the following definition.

**Definition 2.4** ($W_\infty$ distance). Let $\mu, \nu$ be probability distributions over $\mathbb{R}^d$. The infinite Wasserstein distance between $\mu, \nu$ is defined as $W_\infty(\mu, \nu) = \inf_{\gamma \in \Gamma(\mu,\nu)} \mathrm{ess\,sup}_{(X,Y)\sim\gamma} \|X - Y\|$, where $\Gamma(\mu, \nu)$ is the set of all possible coupling of $\mu, \nu$.

## 2.2. Zeroth-Order Optimization

We consider the empirical risk minimization (ERM) problem of minimizing the loss function for a parameter vector $w \in \mathbb{R}^d$ over a given dataset $\mathcal{D} = \{x_i\}_{i=1}^n$:

$$L(w; \mathcal{D}) = \frac{1}{n}\sum_{i=1}^{n}\ell(w; x_i) = \frac{1}{n}\sum_{i=1}^{n}\ell_i(w).$$

To this end, we study a zeroth-order gradient descent (ZOGD) algorithm:

$$w_{t+1} = \Pi_{\mathcal{K}}\left[w_t - \frac{\eta}{K}\sum_{k=1}^{K}\hat{g}_t(w_t; u_{t,k})\right], \text{where } \hat{g}_t(w_t; u_{t,k})$$
$$:= \frac{1}{n}\sum_{i=1}^{n}\mathsf{clip}\left(\frac{\ell_i(w_t + \xi u_{t,k}) - \ell_i(w_t - \xi u_{t,k})}{2\xi}; \Delta\right)u_{t,k},$$
$$(2)$$

Here, $\xi \ge 0$ is the perturbation scale, $\eta > 0$ is the step size, $\mathsf{clip}(y; \Delta)$ denotes the clipping function (defined as$\mathsf{clip}(y; \Delta) \triangleq \frac{y}{\max(1,|y|/\Delta)}$), and the operator $\Pi_{\mathcal{K}}(\cdot)$ denotes projection onto a bounded convex set $\mathcal{K}$ with diameter

---

[2]With a slight abuse of notation, for random variables $X \sim \mu$ and $Y \sim \nu$ we define $D_\alpha(X\|Y) = D_\alpha(\mu\|\nu)$.

$D$. Note that the estimator $\hat{g}_t(w_t; u_{t,k})$ is a two-point approximation of the directional derivative along $u_{t,k}$ and serves as a zeroth-order surrogate for the true gradient $\nabla L(w)$. We refer to this estimator as the *zeroth-order gradient* throughout the paper. The direction vectors $u_{t,k}$ are randomly sampled from an isotropic distribution and will be specified in the algorithm. In the special case $K = 1$, it is common to drawn $u$ from the Euclidean sphere $\mathbb{S}^{d-1}$ uniformly (Zhang et al., 2024a) or multivariate Gaussian distribution (Tang et al., 2024). Our results can be generalized to other zeroth-order optimization methods, such as one-point estimators (Flaxman et al., 2005) and the directional derivative (Nesterov & Spokoiny, 2017).

# 3. Hidden-state DP Analysis for Zeroth-order Optimization

Our goal in this section is to show that the final iterate $w_T$ of the privatized ZOGD algorithm (2) satisfies $(\alpha, \varepsilon(\alpha))$-RDP, with $\varepsilon(\alpha)$ independent of $T$. This can be translated into $(\varepsilon, \delta)$-DP via the standard RDP-to-DP conversion (Mironov, 2017). Equivalently, we aim to upper bound the Rényi divergence between two adjacent output distributions $w_T$ and $w_T'$ corresponding to neighboring datasets $\mathcal{D} \simeq \mathcal{D}'$.

To privatize ZOGD (2), Zhang et al. (2024a) has considered two types of noise injection mechanisms in the case of $K = 1$:

$$(a) : \Pi_{\mathcal{K}} \left[ w_t - \eta \hat{g}_t(w_t; u_t) + \eta G_t^{(1)} u_t \right],$$
$$(b) : \Pi_{\mathcal{K}} \left[ w_t - \eta \hat{g}_t(w_t; u_t) + \eta G_t^{(2)} \right], \quad (3)$$

where $G_t^{(1)} \sim \mathcal{N}(0, \sigma_1^2)$ is scalar Gaussian noise applied in the update direction, and $G_t^{(2)} \sim \mathcal{N}(0, \sigma_2^2 I_d)$ is isotropic Gaussian noise over all coordinates. Under the same privacy budget (i.e., $\sigma_1 = \sigma_2$), Zhang et al. (2024a) showed that mechanism (a) achieves better utility than mechanism (b).

This advantage can be intuitively explained using the composition-based privacy framework: conditioned on the previous iterate $w_{t-1}$, the sensitive information at step $t$ is only leaked through the scalar step size $\hat{g}_t(w_t; u_t)$ along the (random) direction $u_t$. Thus, in mechanism (b), noise injected orthogonally to $u_t$ (i.e., in the remaining $d-1$ dimensions) does not contribute to privacy but still degrades utility. Conversely, mechanism (a) is more efficient but presents technical difficulties: due to the directional nature of the noise, standard shifted-divergence analysis (PABI) techniques are no longer directly applicable. This leads to our central question: *Can we design a noise mechanism that (1) concentrates noise along the update directions to improve utility, while (2) still enabling shifted-divergence analysis for final iterate privacy?*

## 3.1. A Unified Noisy ZOGD mechanism

To address the above, we propose a generalized noise injection scheme that subsumes both (a) and (b) as special cases, while enabling tighter hidden-state privacy analysis:

$$\text{Noisy-ZOGD: } w_{t+1} = \Pi_{\mathcal{B}_R} \Big[ w_t - \frac{\eta}{K} \sum_{k=1}^{K} \hat{g}_t(w_t; u_{t,k})$$
$$+ \frac{\eta}{\sqrt{K}} \sum_{k=1}^{K} G_{t,k}^{(1)} u_{t,k} + \frac{\eta}{\sqrt{d}} G_t^{(2)} \Big], \quad (4)$$

where $\mathcal{B}_R$ is the $\ell_2$ ball of radius $R$ centered at the origin, $G_{t,k}^{(1)} \sim \mathcal{N}(0, \beta_t \sigma^2)$, and $G_t^{(2)} \sim \mathcal{N}(0, (1-\beta_t)\sigma^2 I_d)$. The vectors $\{u_{t,k}\}_{k=1}^{K}$ are orthonormal and drawn uniformly from the Stiefel manifold $V_K(\mathbb{R}^d)$; when $K = 1$, this reduces to the uniform distribution on the sphere, i.e., $u_{t,k} \sim \mathsf{Unif}(\mathbb{S}^{d-1})$.

A couple of remarks are given in order. First, in zeroth-order optimization, using $K > 1$ directions improves gradient estimation (Duchi et al., 2015), though at higher computational cost per iteration. Prior work typically assumes the directions $\{u_{t,k}\}$ are drawn independently from $\mathsf{Unif}(\mathbb{S}^{d-1})$. In contrast, our orthonormal construction facilitates tighter privacy analysis under the hidden-state model. Second, the parameterization in Eq. (4) ensures equivalent total noise variance (and thus utility guarantees) across all $\beta_t \in [0, 1]$ and $K \geq 1$, following the utility analysis of Zhang et al. (2024a) (see Appendix I for details). In particular, setting $K = 1$ with $\beta_t = 1$ and $\beta_t = 0$ recovers mechanisms (a) and (b), respectively. Next, we provide baseline privacy bounds for the above two cases (assuming access to the entire trajectory $\{w_t\}_{t=1}^{T}$) using standard composition.

**Theorem 3.1** (Public-State Privacy Baseline). *The Noisy-ZOGD update rule* (4) *with $T$ iterations satisfies $(\varepsilon, \delta)$-DP, where*

$$\varepsilon = O\left( \sqrt{\frac{\Delta^2 \log(1/\delta) T}{n^2 \sigma^2}} \right) \quad \text{when } \beta_t = 1,$$

$$\varepsilon = O\left( \sqrt{\frac{d \Delta^2 \log(1/\delta) T}{K n^2 \sigma^2}} \right) \quad \text{when } \beta_t = 0. \quad (5)$$

First of all, since $K \leq d$, setting $\beta_t = 0$ always leads to a worse privacy bound under fixed utility constraints. This motivates the design of scalar-directional noise ($\beta_t = 1$), as explored in Zhang et al. (2024a). Second, the above privacy bounds assume that all intermediate iterates $w_1, \ldots, w_T$ are publicly released. As such, the total privacy cost accumulates linearly over iterations, growing unboundedly as $T$ increases. Finally, another way to account the final privacy budget is to treat the mechanism as output perturbation, that is, by ignoring all intermediate noise but only focusing on the last step. Since $w_T \in \mathcal{B}_R$, the Gaussian noise in the

last iterate always ensures an $\left(O\left(\sqrt{R^2\log(1/\delta)/\sigma^2}\right),\delta\right)$-DP guarantee, independent of $T$. However, this bound is generally weaker than the composition-based bound in Theorem 3.1 for most practical $T$.

Our main contribution is a *non-trivial convergent* privacy analysis and guarantee for the final iterate $w_T$. Specifically, for smooth (strongly) convex problems, we show that it satisfies $\left(O\left(\sqrt{\frac{\Delta^2\log(1/\delta)}{n^2\sigma^2}\min(T,\frac{Rn\sqrt{d}}{K\Delta})}\right),\delta\right)$-DP; see Figure 1 for numerical results and comparisons to the standard composition and output perturbation. Note that our analysis extends to non-convex settings, and, while the privacy bound in those cases does not admit a closed-form expression, we provide an numerical privacy accountant to compute the privacy budget. Importantly, our analysis reveals that by carefully tuning $\beta_t \in (0,1)$, one can achieve improved privacy-utility trade-offs using hidden-state analysis. Our results also extend to mini-batch settings and incorporate privacy amplification via subsampling (Balle et al., 2018).

### 3.2. Hidden-state Privacy Analysis for Noisy-ZOGD

Next, we present our main theorem, the first *convergent* privacy bound for the zeroth-order ERM.

**Theorem 3.2** (Hidden-State DP Bound). *Assume $\ell(\cdot,x)$ are $M$-smooth and $\Delta$-Lipschitz for the first argument and all possible data $x$. For any $\alpha > 1, d \geq 2K, \vartheta \geq 0$, define*

$$\bar{c}_1 = \sqrt{1-(1-c^2)\frac{K}{d}+\vartheta(1+c^2)\frac{K}{d}},\ c_2 = \eta M\xi,$$

$$\delta_f = 2(T-\tau)\exp\left(-\frac{3\vartheta^2 dK}{12(d-K)+8\vartheta(d-2K)}\right).$$

$$\rho_\alpha = \sum_{t=\tau}^{T-1}\left(\frac{\alpha}{2\beta_t\sigma^2}\left(\frac{2\Delta}{n}\right)^2 + \frac{\alpha a_t^2 d}{2\eta^2\left(1-\beta_t\right)\sigma^2}\right),$$

$$\rho_\alpha^\star = \min_{\tau,\beta_t,a_t}\rho_\alpha$$

$$s.t.\ \tau\in\{0,\ldots,T-1\},\ \beta_t\in[0,1],\ a_t,z_t\geq 0,$$

$$\forall t\geq\tau, z_{t-1}=\bar{c}_1^{-1}(z_t+a_t-c_2),\ z_T=0,$$

$$z_\tau\geq\min\left(2R,\frac{2\eta\Delta\tau}{\sqrt{K}}\right),$$

*where $c=1+\frac{\eta M}{K}$. If $\ell$ is also convex and choose $\eta \leq 2K/M$, we have $c=1$. If $\ell$ is also $m$-strongly convex and choose $\eta \leq K/M$, we have $c=1-\frac{\eta m}{K}$. Then for any $\delta_p \in (0,1)$, the Noisy-ZOGD update (4) is*

$$\left(\rho_\alpha^\star + \frac{\log(1/\delta_p)}{\alpha-1},\delta_p+\delta_f\right)\text{-DP.}$$

While the tightest privacy bound in Theorem 3.2 requires solving a constrained optimization problem, we emphasize

that any suboptimal feasible solution $\rho_\alpha$ still yields a valid privacy guarantee. That is, any $\tau,\beta_t,a_t$ that satisfy the constraint stated in Theorem 3.2 would result in a valid privacy bound. In Appendix B, we provide numerical results to demonstrate how our bound improves upon standard composition and the naive output perturbation analysis. Although the optimal bound generally lacks a closed-form expression, we can derive a closed-form result in the case of $m$-strongly convex objectives.

**Corollary 3.3** (Closed-from DP guarantee of Noisy-ZOGD). *Assume $\ell(\cdot,x)$ are $M$-smooth, $m$-strongly convex and $\Delta$-Lipschitz for the first argument and all possible data $x$. The Noisy-ZOGD update (4) is $(\varepsilon,\delta)$-DP with the choice $\eta = K/M$, $\max(\frac{20(1+c^2)^2}{3(1-c^2)^2}\log(\frac{4}{\delta}\lceil\frac{MRn\sqrt{2d}}{\Delta}\rceil),1) \leq K \leq d/2$ and $\xi \leq \frac{2\Delta}{n\eta M\sqrt{2d}}$, where*

$$\varepsilon = O\left(\sqrt{\frac{\Delta^2\log(1/\delta)}{n^2\sigma^2}\min(T,\frac{MRn\sqrt{d}}{K\Delta})}\right),$$

$$c = 1-\frac{m}{M}.$$

The bound on $\varepsilon$ in Corollary 3.3 is convergent as $T \to \infty$ and becomes inversely proportional to $K$ in this regime. This behavior contrasts with the standard composition-based privacy bound in Theorem 3.1, where using multiple directions ($K > 1$) offers no privacy gain when $\beta_t = 0$. This distinction highlights a core contribution of our work: our hidden-state analysis reveals that incorporating multiple zeroth-order estimates improves not only utility but also privacy—a phenomenon not observed in analogous first-order methods. Compared to prior results in first-order settings (Altschuler & Talwar, 2022; Chien & Li, 2025), our results demonstrate that using multiple zeroth-order directions provides its own privacy benefits in the zeroth-order regime. While leveraging $K > 1$ directions is computationally more expensive than the standard DP zeroth-order method (i.e., case (a) in (3)), our results unveil its privacy benefit (under fixed utility bound) and thus enable a new privacy-computational complexity tradeoff for DP zeroth-order optimization.

Before presenting the proof, we introduce several technical lemmas and definitions that will be utilized in our hidden-state DP analysis. The first is related to the Lipschitz behavior of the zeroth-order update map.

**Definition 3.4** (Generalized Lipschitz). *For $c_1,c_2 \geq 0$, the mapping $\phi$ is said to be $(c_1,c_2)$-generalized Lipschitz if for any $x,y$ in its domain, $\|\phi(x)-\phi(y)\| \leq c_1\|x-y\| + c_2$.*

We next show that the noiseless ZOGD update $\hat{\psi}_t(w) = w - \frac{\eta}{K}\sum_{k=1}^K \hat{g}_t(w;u_{t,k})$ is indeed generalized Lipschitz with high probability when its first-order counterpart $\phi_t(w) = w - \frac{\eta}{nK}\sum_{i=1}^n \nabla\ell_i(w)$ is Lipschitz. Note that when the

loss is at least smooth, one can show that the GD update is indeed Lipschitz with a proper choice of step size (Hardt et al., 2016).

**Lemma 3.5.** *Let $\hat{\psi}(w) = w - \frac{\eta}{K} \sum_{k=1}^{K} \hat{g}(w; u_k)$, where $\hat{g}$ and $u_k$ are defined in* (4). *Assume the corresponding first order counterpart $\phi$ is $c$-Lipschitz $\ell_i, \ell_i'$ are $\Delta$-Lipschitz and $M$-smooth. Then $\hat{\psi}$ is $(c_1, c_2)$-generalized Lipschitz, where $c_2 = \eta M \xi$ and*

$$c_1 = \sqrt{1 - \sum_{k=1}^{K} \upsilon_k + c^2 \sum_{k=1}^{K} \gamma_k},$$

$$\sum_{k=1}^{K} \upsilon_k \sim \mathsf{Beta}\left(\frac{K}{2}, \frac{d-K}{2}\right),$$

$$\sum_{k=1}^{K} \gamma_k \sim \mathsf{Beta}\left(\frac{K}{2}, \frac{d-K}{2}\right).$$

**Lemma 3.6.** *Let $A, B$ be any unit vector and $u_k$ defined in* (4). *Denoting $\upsilon_k = (\langle A, u_k \rangle)^2$, $\gamma_k = (\langle B, u_k \rangle)^2$ and $c_1 = \sqrt{1 - \sum_{k=1}^{K} \upsilon_k + c^2 \sum_{k=1}^{K} \gamma_k}$. Then for any $\vartheta \geq 0$ and $d \geq 2K$,*

$$\mathbb{P}\left(c_1 \leq \sqrt{1 - (1-c^2)\frac{K}{d} + \vartheta(1+c^2)\frac{K}{d}}\right)$$

$$\geq 1 - 2\exp\left(-\frac{3\vartheta^2 dK}{12(d-K) + 8\vartheta(d-2K)}\right).$$

Note that our Lemma 3.6 is not a global Lipschitz condition. Thus it cannot be applied to the shifted divergence analysis as in the first-order cases (Feldman et al., 2018; Altschuler & Talwar, 2022; Chien & Li, 2025). This is the reason why we explicitly construct an auxiliary process that is "in the middle" of two adjacent process defined below, which allow us to side step the shifted divergence analysis.

Let us introduce the adjacent process of (2) with respect to adjacent datasets $\mathcal{D}, \mathcal{D}'$ are defined as follows

$$w'_{t+1} = \Pi_{\mathcal{B}_R}\left[w'_t - \frac{\eta}{K} \sum_{k=1}^{K} \hat{g}_t(w'_t; u'_{t,k}) \right.$$

$$\left. + \frac{\eta}{\sqrt{K}} \sum_{k=1}^{K} G'^{,(1)}_{t,k} u'_{t,k} + \frac{\eta}{\sqrt{d}} G'^{,(2)}_t\right], \quad (6)$$

where the initializations are identical almost surely $w_0 = w'_0$ and the noises are identically and independently distributed as their counterparts in (4). Finally, we will need to bound the Wasserstein distance of adjacent processes. This is similar to the prior literature about hidden-state DP analysis for first-order method (Chien & Li, 2025; Wei et al., 2024; Chien et al., 2024a).

**Lemma 3.7** (Forward $W_\infty$ tracking). *Let $w_t, w'_t$ be the process defined in* (4) *and* (6).

$$W_\infty(w_t, w'_t) \leq \min\left(2R, \frac{2\eta\Delta t}{\sqrt{K}}\right)$$

Now we are ready to provide the proof of Theorem 3.2.

*Proof.* Let us define the $\hat{\psi}_t(w) = w - \frac{\eta}{K} \sum_{k=1}^{K} \hat{g}_t(w; u_{t,k})$ to be the zeroth-order update map and $\phi_t(w) = w - \frac{\eta}{nK} \sum_{i=1}^{n} \nabla \ell_i(w)$ be the corresponding first order update. Consider any two adjacent processes $W_t, W'_t$ defined in (4), (6), we can construct a specific coupling as follows.

$$W_{t+1} \stackrel{d}{=} \Pi_{\mathcal{B}_R}\left[\hat{\psi}_t(W_t) + Y_t + Z_t\right],$$

$$W'_{t+1} \stackrel{d}{=} \Pi_{\mathcal{B}_R}\left[\hat{\psi}'_t(W'_t) + Y_t + Z_t\right],$$

where $Z_t \sim N(0, \sigma^2_{2,t}I_d)$, $Y_t \stackrel{d}{=} \sum_{k=1}^{K} G_{t,k}u_{t,k}$, $G_{t,k} \sim N(0, \sigma^2_{1,t})$. Now we can construct a specific auxiliary process $\widetilde{W}_t$ depending on $\mathcal{D}, \mathcal{D}'$ as follows: for all $t > \tau$,

$$\widetilde{W}_{t+1} \stackrel{d}{=} \Pi_{\mathcal{B}_R}\left[\hat{\psi}'_t(\widetilde{W}_t) + Y_t + b_t + Z_t + v_t\right],$$

where $\widetilde{W}_t = W'_t$ for all $t \leq \tau$, and $b_t, v_t$ are defined as follows:

$$b_t := \hat{\psi}_t(\widetilde{W}_t) - \hat{\psi}'_t(\widetilde{W}_t), \ d_t := \hat{\psi}_t(W_t) - \hat{\psi}_t(\widetilde{W}_t)$$

$$v_t := \min(a_t, (\|d_t\| - z_{t+1})_+)\frac{d_t}{\|d_t\|}, \ \text{for } d_t \neq 0,$$

$v_t = 0$ otherwise. Note that we have $\sigma^2_{1,t} = \frac{\eta^2}{K}\beta_t\sigma^2$ and $\sigma^2_{2,t} = \frac{\eta^2}{d}(1-\beta_t)\sigma^2$ to match the original Noisy-ZOGD update (4). Also note that $\ell_i = \ell_i'$ except for one index according for any pair of adjacent processes. Several remarks for $\widetilde{W}_t$. First, $b_t$ contains scalar differences on directions $u_{t,k}$. This part will be privatized by the non-isotropic noise $Y_t$. Second, $v_t$ is a general vector differences, but our contruction ensures that $\|v_t\| \leq a_t$ always.

First, note that by Lemma 3.7, we know that at time $t = \tau$ we have

$$\|W_\tau - W'_\tau\| \leq \min\left(2R, \frac{2\eta\Delta\tau}{\sqrt{K}}\right) \leq z_\tau.$$

Next we will bound the TV distance between $W_T, \widetilde{W}_T$. Note that by definition of $b_t$, we have

$$\widetilde{W}_{t+1} \stackrel{d}{=} \Pi_{\mathcal{B}_R}\left[\hat{\psi}'_t(\widetilde{W}_t) + Y_t + b_t + Z_t + v_t\right]$$

$$= \Pi_{\mathcal{B}_R}\left[\hat{\psi}_t(\widetilde{W}_t) + Y_t + Z_t + v_t\right],$$

which is the same update as $W_t$ except for the shift $v_t$. Next we define the event

$$\mathcal{G}_t = \{\|\hat{\psi}_t(W_t) - \hat{\psi}_t(\widetilde{W}_t)\| \leq \bar{c}_1\|W_t - \widetilde{W}_t\| + c_2\}. \quad (7)$$

By Lemma 3.6, we know that

$$\mathbb{P}(\mathcal{G}_t|\mathcal{F}_t) \leq \frac{\delta_f}{T - \tau},$$

where $\mathcal{F}_t$ is the sigma-field generated by all randomness up to time $t$ in our coupled construction. Then we also define

$$\mathcal{H}_t = \{\|W_t - \widetilde{W}_t\| \leq z_t\}. \quad (8)$$

We already know that $\mathcal{H}_\tau$ holds almost surely by Lemma 3.7 and our constuction $\widetilde{W}_\tau = W'_\tau$. Then under $\mathcal{H}_t \cap \mathcal{G}_t$,

$$\|d_t\| \leq \bar{c}_1 z_t + c_2 \leq z_{t+1} + a_t \quad (9)$$

by the feasibility condition of $z_t, a_t$. Then by construction of $v_t$ and the projection $\Pi_{\mathcal{B}_R}$ is 1-Lipschitz, this implies

$$\|W_{t+1} - \widetilde{W}_{t+1}\| \leq \|d_t - v_t\| \leq z_{t+1}. \quad (10)$$

Thus $\mathcal{H}_t \cap \mathcal{G}_t \subseteq \mathcal{H}_{t+1}$. Inductively, $\cap_{s=\tau}^t \mathcal{G}_s \subseteq \mathcal{H}_{t+1}$ for all $t \geq \tau$. In particular, on $\mathcal{G} := \cap_{s=\tau}^{T-1}\mathcal{G}_s$, we have $\mathcal{H}_T$. Since $z_T = 0$, this implies $W_T = \widetilde{W}_T$ on $\mathcal{G}$. Moreover, by union bound we have

$$\mathbb{P}(\mathcal{G}^c) \leq \sum_{t=\tau}^{T-1} \mathbb{P}(\mathcal{G}_t^c) \leq \delta_f. \quad (11)$$

Therefore, by the coupling inequality,

$$TV(W_T, \widetilde{W}_T) \leq \mathbb{P}(W_T \neq \widetilde{W}_T) \leq \delta_f. \quad (12)$$

This complete the first half of the proof. Next we will bound the Rényi divergence among $\widetilde{W}_T, W'_T$.

At time $t \geq \tau$, condition on the past, we know that $\widetilde{W}_t, W'_t$ differs only at shifts $b_t, v_t$ at each step. For $b_t$ part, it is handled by the scalar Gaussian in the $K$-dimensional basis coordinate $u_{t,k}$ ($Y_t$), where the corresponding sensitivity is $\|b_t\|$. Similarly, $v_t$ is handled by the isotropic Gaussian $Z_t$, where the corresponding sensitivity is $\|v_t\|$.

For the $b_t, Y_t$ part, condition on the past we have

$$\|b_t\|^2 = \|\hat{\psi}_t(w) - \hat{\psi}'_t(w)\|^2 \leq \frac{\eta^2}{K}(\frac{2\Delta}{n})^2, \quad (13)$$

since at each coordinate $u_{t,k}$ the sensitivity is $\frac{2\Delta}{n}$ and the $K$ coordinate basis $u_{t,k}$ are orthonormal. As a result, the corresponding Rényi privacy cost is

$$\frac{\alpha\|b_t\|^2}{2(\eta^2\beta_t\sigma^2/K)} \leq \frac{\alpha}{2\beta_t\sigma^2}(\frac{2\Delta}{n})^2 \quad (14)$$

For the $v_t, Z_t$ part, condition on the past we have

$$\|v_t\|^2 \leq a_t^2 \quad (15)$$

by construction of $v_t$. Thus, the corresponding Rényi privacy cost is

$$\frac{\alpha\|v_t\|^2}{2(\eta^2(1 - \beta_t)\sigma^2/d)} \leq \frac{\alpha da_t^2}{2\eta^2(1 - \beta_t)\sigma^2}. \quad (16)$$

Moreover, $\widetilde{W}_\tau = W'_\tau$ by our construction. Thus, by strong composition theorem (Lemma 2.3) to the full adaptive Gaussian transcript, and then post-processing property (Lemma 2.2), we have

$$D_\alpha(\widetilde{W}_t\|W'_t) \leq \rho_\alpha. \quad (17)$$

Note that the same argument hold for $D_\alpha(W'_t\|\widetilde{W}_t)$ by symmetry. Finally, we can convert it to DP bound by RDP-DP conversion (Mironov, 2017). Specifically, for fixed $\delta_p \in (0, 1)$, and every measurable output set $S$, (17) implies

$$\mathbb{P}(\widetilde{W}_T \in S) \leq \exp(\rho_\alpha + \frac{\log(1/\delta_p)}{\alpha - 1})\mathbb{P}(W'_T \in S) + \delta_p.$$

Also, the TV distance result (12) implies

$$\mathbb{P}(W_T \in S) \leq \mathbb{P}(\widetilde{W}_T \in S) + \delta_f. \quad (18)$$

Combining the inequalities gives the desired DP inequality. Note that the same argument holds with $D, D'$ exchanged, giving the reverse DP inequality by symmetry of replacement DP. Together we complete the proof.

$$\square$$

**Further Remarks.** There are several important observations regarding our results and analysis. First, although our main analysis focuses on the full-batch setting, Theorem 3.2 naturally extends to the mini-batch regime. In particular, we present results for the case of sampling without replacement, where we replace the Rényi privacy cost of Gaussian mechanism for the $b_t, Y_t$ part with Sampled Gaussian mechanism (Mironov et al., 2019). The corresponding privacy bounds and analysis are provided in Appendix H. While it is also possible to extend our results to other mini-batch schemes, such as shuffled cyclic mini-batches (Ye & Shokri, 2022; Chien & Li, 2025), we leave such generalizations for future work.

Second, a key analysis-driven design choice in our method is the selection of update directions $\{u_{t,k}\}$. While standard zeroth-order optimization methods (Duchi et al., 2015; Liu et al., 2024a) often sample directions independently (i.i.d.), such a choice would significantly complicate the analysis, particularly the concentration bounds involving the term $c_1$ in Lemma 3.6, and result in weaker privacy guarantees. We

discuss the implications of the i.i.d. sampling scheme in Appendix K. Our use of orthonormal directions enables the application of tight quantile bounds from the beta distribution, which are critical to our privacy analysis. Notably, the privacy benefit of using orthonormal $u_{t,k}$ directions appears to be novel in the literature; under the standard composition-based analysis, orthogonality does not influence the privacy guarantee and is therefore overlooked.

Third, our Lemma 3.6 provides a point-wise Lipschitz bound on a high probability event. One may attempt to combine it with the shifted-divergence analysis, specifically the shifted-reduction lemma (Feldman et al., 2018; Altschuler & Talwar, 2022). Unfortunately, such a combination is invalid as the shifted-reduction lemma requires a global Lipschitz bound (i.e., almost surely). To alleviate the issue, we instead directly construct an auxiliary process to bridge $W_T, W_T'$ and side-step the need for shift-divergence analysis. Our novel construction may be of independent interest.

Finally, some of the assumptions in our analysis can likely be relaxed. For instance, the requirement that each loss function $\ell_i$ is both Lipschitz and $M$-smooth simplifies the analysis but is not strictly necessary. Relaxing the Lipschitz assumption, for example, mainly affects the proof that the (noiseless) gradient update map $\phi$ is Lipschitz, a property that is crucial for the shifted divergence analysis. Recent work by Kong & Ribero (2024) has shown that the clipped first-order gradient map remains Lipschitz even without assuming Lipschitz continuity of the loss itself. Additionally, Chien & Li (2025) demonstrated that the smoothness requirement can be weakened to Hölder continuity of the gradient in the context of shifted divergence for first-order methods. We leave the extension of our analysis to these broader settings as an important direction for future work.

## 4. Related Works

**Hidden-state Noisy-SGD.** The hidden-state privacy analysis for the first-order optimization method, specifically SGD, has raised great attention recently. (Chourasia et al., 2021; Ye & Shokri, 2022) leverage the analysis of Langevin dynamics for DP guarantees but require strong convexity on the loss. (Chien et al., 2024b) adopts a similar analysis but for the machine unlearning problem. (Asoodeh & Diaz, 2023) establishes the convergent DP bound by establishing the contraction of the hockey stick divergence of the Noisy-SGD process. (Feldman et al., 2018; Altschuler & Talwar, 2022; Altschuler et al., 2024) derive convergent DP bound for Noisy-SGD based on formalizing shifted Rényi divergence analysis, albeit the loss is required to be convex. (Chien et al., 2024a) adopted their analysis for establishing privacy guarantees for the machine unlearning problem and (Chien & Li, 2025) relaxed the convex and smoothness assumption in these works. Recently, many

works have also studied the hidden-state privacy for Noisy-SGD by either establishing DP guarantees or performing privacy auditing (Kong & Ribero, 2024; Annamalai, 2024; Cebere et al., 2024; Steinke et al., 2024). All of these works focus on the first-order optimization method, while we study the hidden-state DP guarantee for the zeroth-order method. Moreover, as we discussed a direct generalization of the shifted-divergence analysis fails. Our analysis is based on a novel construction of an auxiliary process that side-steps the need for shifted-divergence analysis.

**Differentially Private Zeroth-order Optimization.** Zeroth-order optimization under DP constraint has only been studied very recently but gain increasing attention. Zhang et al. (2024b) is the first to study DP zeroth-order optimization. Zhang et al. (2024a) improves their results by observing the benefit of scalar Gaussian noise (see our discussion around Theorem 3.1) and proposed the state-of-the-art algorithm DPZero, albeit they focus on the case $K = 1$. Concurrently, Tang et al. (2024) proposed a method similar to DPZero but with the update direction $u$ being drawn from standard normal. They demonstrate the benefit of DP zeroth-order optimization in fine-tuning the Large Languague Model (LLM) but without providing utility analysis. Liu et al. (2024b) propose a stage-wise zeroth-order optimization DP-ZOSO building upon DPZero and demonstrates empirical gain over DPZero. All of these works do not establish the convergent privacy loss under hidden-state assumption as in our work. Our work provides a non-trivial convergent privacy loss for zeroth-order optimization under DP constraint over the bounded domain. We also elucidate the privacy benefit of leveraging $K \geq 1$ zeroth-order gradient and choosing orthonormal update direction $u$, which was not studied in the prior literature as well to the best of our knowledge. Very recently, Bao et al. (2025) also studied the benefit of leveraging $K$ (but i.i.d.) update directions in zeroth-order optimization under a public state setting. It is interesting to see if their analysis can be combined with our results.

Gupta et al. (2024) investigates whether zeroth-order methods can offer inherent privacy purely from the randomness of the update directions, **without any additive noise.** Their main result shows that this is not sufficient to guarantee differential privacy, highlighting the necessity of explicit noise addition. This observation is fully consistent with our work and, in fact, further reinforces the relevance of our contribution. Indeed, Gupta et al. (2024) also stated the following open problem: "Although zeroth-order estimators do not offer inherent privacy on their own, can they amplify the privacy of other (additive) noisy methods?". Interestingly, our work provides a positive answer to this open question, where we show that the randomness in the update directions is indeed helpful when combined with additive Gaussian noise (under hidden-state setting).

## Acknowledgements

EC and PL are partially supported by NSF awards PHY-2117997, IIS-223956, CCF-2402816, and JPMC faculty award. EC is also partially supported by the Yushan Young Fellow Program (115V1070-1) from the Ministry of Education, Taiwan, and NTU Artificial Intelligence Center of Research Excellence (114M7069-01) within Taiwan Centers of Excellence in Artificial Intelligence.

## Impact Statement

This paper presents work whose goal is to advance the field of machine learning privacy. There are many potential societal consequences of our work, none of which we feel must be specifically highlighted here.

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

# A. Conclusion and future works

We provide the first convergent DP bound under the hidden-state assumption for smooth losses. Our analysis elucidates several privacy-beneficial designs in the zeroth-order method that cannot be demonstrated via the standard DP composition theorem. Our work paves the way to advanced DP zeroth-order approaches, which we believe to achieve a better privacy-utility trade-off in large language model fine-tuning under DP and memory constraints in the future.

**Future directions.** While we have established the first convergent DP guarantee for the ZOGD method, it is not ready to be directly applied to LLM fine-tuning. It is also unclear at this moment that, in practice, what type of learning task does a $\varepsilon \le 1$ lead to a parameter setting with reasonable complexity and utility. This theory-practice gap is shared by prior theoretical works on hidden-state analysis on first-order methods (Altschuler & Talwar, 2022; Altschuler et al., 2024; Ye & Shokri, 2022; Chien & Li, 2025), as we all need several structural assumptions (such as smoothness) to establish such results. Furthermore, as discussed in the main text, we left several generalizations for future work. This includes using shuffled cyclic minibatch (Ye & Shokri, 2022), relaxing the Lipschitz loss assumption using the analysis of (Kong & Ribero, 2024), and relaxing the smoothness assumption using the analysis of (Chien & Li, 2025). Improvements along these directions will reduce the aforementioned gap of DP zeroth-order methods' applicability to modern LLM training.

# B. Details about numerical results

**Setting for Figure 1.** We report the privacy loss of Noisy-ZOGD (4) under different privacy analysis. We set the radius of the projected set to be $R = 1$, smooth constant $M = 1$, strongly convex constant $m = 0.9$, and the clipped norm $\Delta = 1$. For a fair comparison, we first compute the RDP budget for each analysis and then convert it to DP by the standard conversion (Mironov, 2017). For output perturbation, the RDP loss under this setting is $\frac{\alpha(2R)^2 d}{2\eta^2\sigma^2}$. Note that since $\eta = K/M$, for simplicity, we use the latest tested $\eta$, which gives the smallest privacy loss for output perturbation. For standard composition theorem, after RDP to DP conversion, it is basically the bound stated in Theorem 3.1 with $\beta_t = 1$. For our approach, we adopted the same simplification as in the proof of Corollary 3.3 that applies to Theorem 3.2 for strongly convex problems and kept all constants explicit. We also provide results for the nonconvex case in Figure 2.

# C. Proof of Theorem 3.1

The proof is quite standard in the DP literature (i.e., based on the analysis of (Mironov, 2017)). We state it here for completeness.

*Proof.* **Case $\beta_t = 1$:** Note that in this case, on all direction $u_{t,k}$ we apply the Gaussian mechanism result for each step $t$. The sensitivity of each direction $u_{t,k}$ is

$$\|\frac{1}{K}\hat{g}_t(w; u_{t,k})\| = \frac{\Delta}{nK}. \tag{19}$$

This is due to the clipping operation. As a result, for each direction Gaussian mechanism gives the privacy loss (in RDP) of

$$\frac{\alpha\Delta^2/(nK)^2}{2\sigma^2/K} = \frac{\alpha\Delta^2}{2n^2K\sigma^2}. \tag{20}$$

Then we apply the composition theorem (Lemma 2.3) over $K$ direction and $T$ iterations, followed by the RDP-DP conversion (Mironov, 2017), we have the desired result.

**Case $\beta_t = 0$:** In this case, the one-step sensitivity is the norm of the full zeroth-order gradient, which is

$$\|\frac{1}{K}\sum_k \hat{g}_t(w; u_{t,k})\|^2 \overset{(a)}{=} \frac{1}{K}\|\frac{\Delta}{n}\|^2 = \frac{\Delta^2}{n^2K}, \tag{21}$$

where (a) is due to the orthonormal choice of $u_{t,k}$. Apparently, using a non-orthonormal choice (i.e., i.i.d. directions) only leads to worse sensitivity. As a result, the Gaussian mechanism gives the privacy loss (in RDP) of

$$\frac{\alpha\Delta^2/(n^2K)}{2\sigma^2/d} = \frac{\alpha\Delta^2 d}{2n^2K\sigma^2}. \tag{22}$$

Then we apply the composition theorem (Lemma 2.3) over $T$ iterations, followed by the RDP-DP conversion (Mironov, 2017), we have the desired result. Together we complete the proof. $\square$

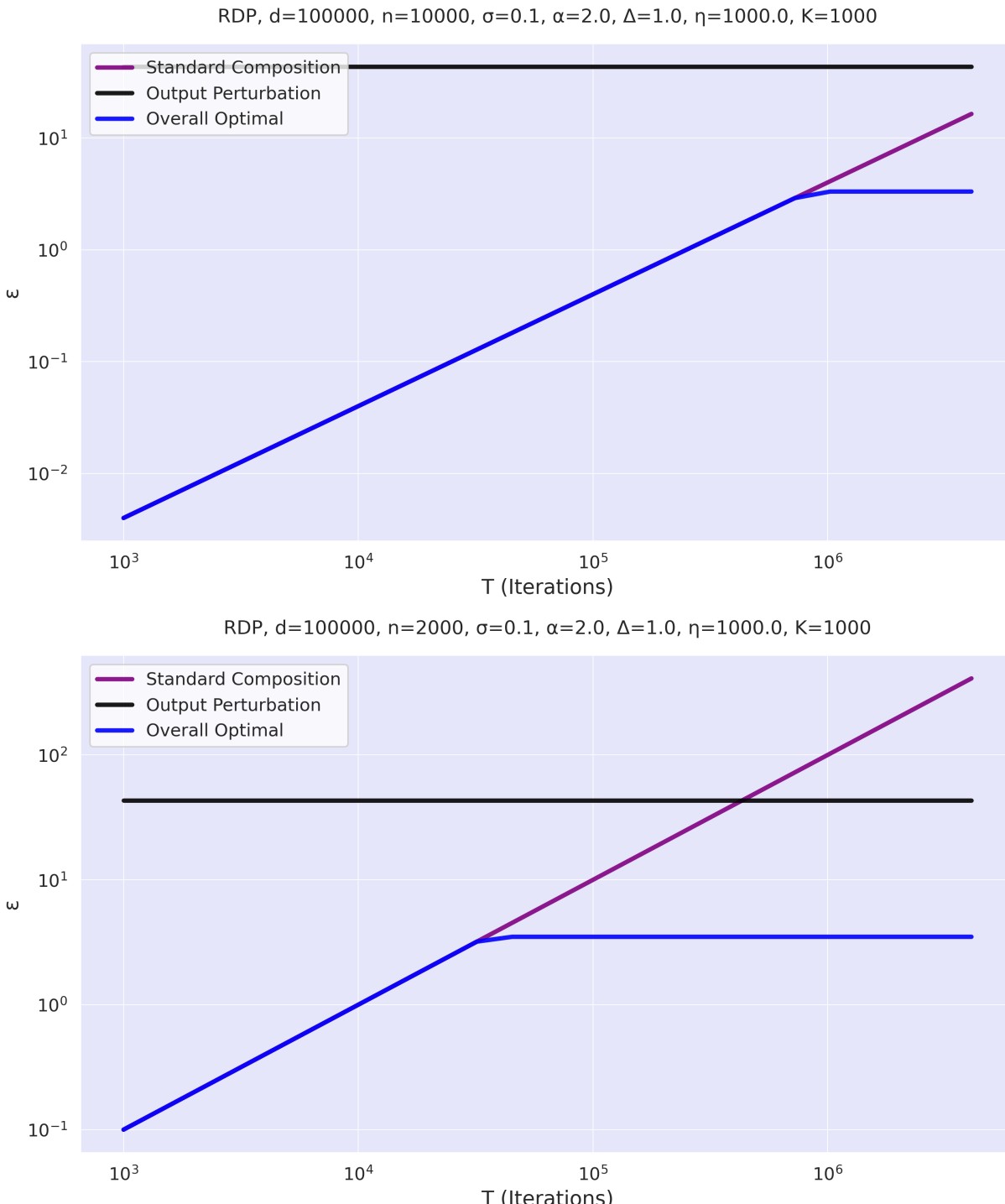

*Figure 2.* The DP guarantees for smooth (and not necessarily convex) losses over the bounded domain versus training iteration $T$. We compared two different sample sizes $n = 1000$ and $n = 2000$ under the fixed noise variance $\sigma^2$, model size $d$, zeroth order dim $K$. See Theorem 3.2 of the explicit bound.

# D. Proof of Corollary 3.3

*Corollary.* Assume $\ell(\cdot, x)$ are $M$-smooth, $m$-strongly convex and $\Delta$-Lipschitz for the first argument and all possible data $x$. The Noisy-ZOGD update (4) is $(\varepsilon, \delta)$-DP with the choice $\eta = K/M$, $\max(\frac{20(1+c^2)^2}{3(1-c^2)^2} \log(\frac{4}{\delta} \lceil \frac{Rn\sqrt{2d}}{\eta\Delta} \rceil), 1) \leq K \leq d/2$ and $\xi \leq \frac{2\Delta}{n\eta M\sqrt{2d}}$, where

$$\varepsilon = O\left( \sqrt{\frac{\Delta^2 \log(1/\delta)}{n^2\sigma^2}} \min(T, \frac{MRn\sqrt{d}}{K\Delta}) \right), \quad c = 1 - \frac{m}{M}. \tag{23}$$

*Proof.* This corollary is a direct application of our main Theorem 3.2. While the optimization stated in Theorem 3.2 does not have a close form solution in general, we can still obtain a simplified bound in the special case. Specifically, we hope to have $c_1 = 1$, where the sufficient condition is

$$1 - (1 - c^2)\frac{K}{d} + \vartheta(1 + c^2)\frac{K}{d} = 1 \Leftrightarrow 0 \leq \vartheta = \frac{1 - c^2}{1 + c^2}. \tag{24}$$

As a result, we will choose $\vartheta = \frac{1-c^2}{1+c^2}$ so that $c_1 = 1$. Note that in order to have $\vartheta > 0$, we need $c < 1$. This is important since it will directly impact the error probability $\delta_f$ in the exponent. In the meanwhile, the Lipschitz constant $c$ of the first order gradient update map $\phi$ is as follows: If $\ell_i$ are $M$-smooth and $m$-strongly convex, then if $\frac{\eta}{K} \leq \frac{1}{M}$ we have $c = 1 - \frac{\eta m}{K}$. With these choices and setup, the bound in Theorem 3.2 for any $\tau, \beta_t, a_t$ becomes

$$\rho_\alpha = \sum_{t=\tau}^{T-1} \frac{\alpha}{2\beta_t\sigma^2}(\frac{2\Delta}{n})^2 + \sum_{t=\tau}^{T-1} \frac{\alpha a_t^2 d}{2\eta^2(1-\beta_t)\sigma^2}, \tag{25}$$

$$\text{where } a_t, z_t \geq 0 \; \forall t \geq \tau, \; z_{t-1} = z_t + a_t - c_2, \; z_T = 0, \; z_\tau \geq \min(2R, \frac{2C\tau}{\sqrt{K}}), \tag{26}$$

$$c_1 = 1, \; c_2 = \eta M\xi, \; \mathbb{P}(\mathcal{G}) \geq 1 - \delta_f. \tag{27}$$

By rolling out the recursion $z_{t-1} = z_t + a_t - c_2$ and combining the constraint on $z_t$ with those of $a_t$, we have

$$\rho_\alpha \leq \sum_{t=\tau}^{T-1} \frac{\alpha}{2\beta_t\sigma^2}(\frac{2\Delta}{n})^2 + \sum_{t=\tau}^{T-1} \frac{\alpha a_t^2 d}{2\eta^2(1-\beta_t)\sigma^2}, \tag{28}$$

$$\text{where } a_t \geq c_2 \; \forall t \geq \tau, \; \sum_{t=\tau}^{T-1}(a_t - c_2) \geq \min(2R, \frac{2C\tau}{\sqrt{K}}). \tag{29}$$

Now let us set $a_t' = a_t - c_2$, then we have

$$\rho_\alpha \leq \sum_{t=\tau}^{T-1} \frac{\alpha}{2\beta_t\sigma^2}(\frac{2\Delta}{n})^2 + \sum_{t=\tau}^{T-1} \frac{\alpha(a_t' + c_2)^2 d}{2\eta^2(1-\beta_t)\sigma^2}, \tag{30}$$

$$\text{where } a_t' \geq 0 \; \forall t \geq \tau, \; \sum_{t=\tau}^{T-1} a_t' \geq \min(2R, \frac{2C\tau}{\sqrt{K}}). \tag{31}$$

Then we can adopt the elementary inequality $(a_t' - c_2)^2 \leq 2(a_t')^2 + 2c_2^2$, which leads to

$$\rho_\alpha \le \sum_{t=\tau}^{T-1} \frac{\alpha}{2\beta_t \sigma^2} (\frac{2\Delta}{n})^2 + \sum_{t=\tau}^{T-1} \frac{2\alpha((a_t')^2 + c_2^2)d}{2\eta^2(1-\beta_t)\sigma^2}, \tag{32}$$

$$\text{where } a_t' \ge 0 \ \forall t \ge \tau, \ \sum_{t=\tau}^{T-1} a_t' \ge \min(2R, \frac{2C\tau}{\sqrt{K}}). \tag{33}$$

By Cauchy-Schwartz as in (Chien & Li, 2025), we have that

$$\left(\sum_{t=\tau}^{T-1} \frac{(a_t')^2}{1-\beta_t}\right) \cdot \left(\sum_{t=\tau}^{T-1} (1-\beta_t)\right) \ge \left(\sum_{t=\tau}^{T-1} a_t'\right)^2 \ge \left(\min(2R, \frac{2C\tau}{\sqrt{K}})\right)^2, \tag{34}$$

where the last inequality is by our constraint $\sum_{t=\tau}^{T-1} a_t' \ge \min(2R, \frac{2C\tau}{\sqrt{K}})$. Apparently, both of the equalities are attainable for some $a_t' \ge 0$. We can choose such a solution of $a_t'$ and plug them in our bound. Additionally, for simplicity we choose $\beta_t = 1/2$. Together we have

$$\rho_\alpha \le (T-\tau)\left(\frac{\alpha}{\sigma^2}(\frac{2\Delta}{n})^2 + \frac{2\alpha c_2^2 d}{\eta^2 \sigma^2}\right) + \frac{4\alpha \left(\min(2R, \frac{2C\tau}{\sqrt{K}})\right)^2 d}{(T-\tau)\eta^2 \sigma^2}. \tag{35}$$

Recall that $c_2 = \eta M \xi$. When we choose $\xi \le \frac{2\Delta}{n\eta M \sqrt{2d}}$, we have $\left(\frac{\alpha}{\sigma^2}(\frac{2\Delta}{n})^2 + \frac{2\alpha c_2^2 d}{\eta^2 \sigma^2}\right) \le \frac{2\alpha}{\sigma^2}(\frac{2\Delta}{n})^2$. Thus the bound can be further simplified as

$$\rho_\alpha \le (T-\tau)\frac{2\alpha}{\sigma^2}(\frac{2\Delta}{n})^2 + \frac{4\alpha \left(\min(2R, \frac{2C\tau}{\sqrt{K}})\right)^2 d}{(T-\tau)\eta^2 \sigma^2} \tag{36}$$

$$\le (T-\tau)\frac{2\alpha}{\sigma^2}(\frac{2\Delta}{n})^2 + \frac{4\alpha (2R)^2 d}{(T-\tau)\eta^2 \sigma^2}. \tag{37}$$

Now we can try to choose the best $\tau$ to optimize the bound. The optimal integer $\tau$ should satisfy

$$T - \tau = \lceil \frac{nR\sqrt{2d}}{\Delta\eta} \rceil \tag{38}$$

Apparently, this choice is possible for sufficiently large $T \ge \lceil \frac{nR\sqrt{2d}}{\Delta\eta} \rceil$. We will discuss the bound for $T < \lceil \frac{nR\sqrt{2d}}{\Delta\eta} \rceil$ later. By plugging in this choice and assuming $\lceil \frac{nR\sqrt{2d}}{\Delta\eta} \rceil = \frac{nR\sqrt{2d}}{\Delta\eta}$, our bound becomes

$$\frac{8\alpha\Delta R\sqrt{2d}}{\eta n \sigma^2} \tag{39}$$

By RDP-DP conversion (Mironov, 2017), we can optimize $\alpha$ for the $(\varepsilon, \frac{\delta}{2})$-DP guarantee, where we reserve the other $\delta/2$ for the error probability $\delta_f$. The resulting $\varepsilon$ will be

$$\varepsilon = O\left(\sqrt{\frac{M\Delta R\sqrt{2d}\log(1/\delta)}{K\eta n \sigma^2}}\right), \tag{40}$$

where we recall that $\eta = K/M$. On the other hand, note that we still have to deal with the error probability regarding $c_1$. With the choice of $\eta = \frac{K}{M}$, thus we have $c = 1 - \frac{m}{M}$. This implies that we should choose $\vartheta = \frac{1-c^2}{1+c^2}$ to ensure $c_1 = 1$ while keeping $\vartheta$ the smallest possible. The corresponding error probability bound can be simplified as

$$2(T - \tau) \exp\left(-\frac{3\vartheta^2 K}{20}\right) = 2\lceil\frac{Rn\sqrt{2d}}{\eta\Delta}\rceil \exp\left(-(\frac{1-c^2}{1+c^2})^2\frac{3K}{20}\right), \tag{41}$$

We need this error probability to be upper bounded by $\delta/2$. This is possible when we choose $K$ satisfying

$$K \geq (\frac{1-c^2}{1+c^2})^{-2}\frac{20}{3}\log(\frac{4}{\delta}\lceil\frac{Rn\sqrt{2d}}{\eta\Delta}\rceil), \ \eta = \frac{K}{M}. \tag{42}$$

This completes the proof for the convergent part. For the case $T < \lceil\frac{Rn\sqrt{2d}}{\eta\Delta}\rceil$, we can simply choose $\tau = 0$ and then $\beta_t = 1$, where the bound in stated in the Theorem 3.2 becomes

$$\frac{\alpha T}{2\sigma^2}(\frac{2\Delta}{n})^2. \tag{43}$$

Together we complete the proof. $\qquad\square$

# E. Proof of Lemma 3.5

**Lemma.** *Consider the setting $\hat{\psi}_t, u_{t,k}$ defined in this section. Assume the corresponding first order counterpart $\phi$ is $c$-Lipschitz for $\ell_i, \ell_i'$ being $\Delta$-Lipschitz and $M$-smooth. Then $\hat{\psi}_t$ is $(c_1, c_2)$-generalized Lipschitz, where*

$$c_1 = \sqrt{1 - \sum_{k=1}^{K} \upsilon_k + c^2 \sum_{k=1}^{K} \gamma_k}, \ c_2 = \eta M\xi, \tag{44}$$

$$\sum_{k=1}^{K} \upsilon_k \sim Beta(\frac{K}{2}, \frac{d-K}{2}), \ \sum_{k=1}^{K} \gamma_k \sim Beta(\frac{K}{2}, \frac{d-K}{2}). \tag{45}$$

*Proof.* Note if $\ell$ is $M$-smooth, then for any $w, u$ we have

$$\ell(w + \xi u) = \ell(w) + \xi \cdot \langle\nabla\ell(w), u\rangle + \underbrace{(\ell(w + \xi u) - (\ell(w) + \xi \cdot \langle\nabla\ell(w), u\rangle))}_{\triangleq\zeta(w,u,\xi)\leq\frac{1}{2}M\xi^2}.$$

By the assumption that $\ell_i, \ell_i'$ are $\Delta$-Lipschitz, we can safely remove the clipping. These imply that

$$\hat{g}_t(w) = \frac{1}{nK}\sum_{i=1}^{n}\sum_{k=1}^{K}\frac{\ell_i(w + \xi u_{t,k}) - \ell_i(w - \xi u_{t,k})}{2\xi}u_{t,k} \tag{46}$$

$$= \frac{1}{nK}\sum_{i=1}^{n}\sum_{k=1}^{K}\frac{2\xi\langle\nabla\ell_i(w), u_{t,k}\rangle + \zeta(w, u_{t,k}, \xi) - \zeta(w, -u_{t,k}, \xi)}{2\xi}u_{t,k} \tag{47}$$

$$= \frac{1}{nK}\sum_{i=1}^{n}\sum_{k=1}^{K}\langle\nabla\ell_i(w), u_{t,k}\rangle u_{t,k} + \frac{\zeta(w, u_{t,k}, \xi) - \zeta(w, -u_{t,k}, \xi)}{2\xi}u_{t,k} \tag{48}$$

As a result, we have

$$\|\hat{\psi}_t(x) - \hat{\psi}_t(y)\| = \|(x - y) - \eta(\hat{g}_t(x) - \hat{g}_t(y))\| \tag{49}$$

$$\overset{(a)}{\leq} \|(x - y) - \eta\left(\frac{1}{nK}\sum_{i=1}^{n}\sum_{k=1}^{K}\langle\nabla\ell_i(x) - \nabla\ell_i(y), u_{t,k}\rangle u_{t,k}\right)\| + \eta M\xi. \tag{50}$$

where (a) is due to $|\zeta(w, u, \xi)| \leq \frac{1}{2} M \xi^2$ by $M$-smoothness of $\ell_i$, $u_{t,k}$ being unit norm and triangle inequality. Next we denote $\frac{1}{n} \sum_{i=1}^{n} \ell_i(w) = \bar{\ell}(w)$ and denote $\tilde{\eta} = \frac{\eta}{K}$. Then we can proceed as follows

$$\|(x-y) - \tilde{\eta} \sum_{k=1}^{K} \langle \nabla \bar{\ell}(x) - \nabla \bar{\ell}(y), u_{t,k} \rangle u_{t,k} \|^2 \tag{51}$$

$$\stackrel{(a)}{=} \|(x-y) - \sum_{k=1}^{K} \langle x-y, u_{t,k} \rangle u_{t,k}\|^2 + \|\sum_{k=1}^{K} \langle x-y, u_{t,k} \rangle u_{t,k} - \tilde{\eta} \sum_{k=1}^{K} \langle \nabla \bar{\ell}(x) - \nabla \bar{\ell}(y), u_{t,k} \rangle u_{t,k}\|^2$$

$$= \|(x-y) - \sum_{k=1}^{K} \langle x-y, u_{t,k} \rangle u_{t,k}\|^2 + \|\sum_{k=1}^{K} \langle \phi(x) - \phi(y), u_{t,k} \rangle u_{t,k}\|^2$$

$$\stackrel{(b)}{=} (1 - \sum_{k=1}^{K} \upsilon_k)\|x-y\|^2 + (\sum_{k=1}^{K} \gamma_k)\|\phi(x) - \phi(y)\|^2$$

$$\stackrel{(c)}{\leq} (1 - \sum_{k=1}^{K} \upsilon_k + c^2 \sum_{k=1}^{K} \gamma_k)\|x-y\|^2,$$

where $\upsilon_k = (\langle \frac{x-y}{\|x-y\|}, u_{t,k} \rangle)^2$, $\gamma_k = (\langle \frac{\phi(x)-\phi(y)}{\|\phi(x)-\phi(y)\|}, u_{t,k} \rangle)^2$. (a) is due to the subspace of the first term being orthogonal to the subspace of the second term. (b) is due to the fact that $u_{t,k}$ are orthonormal. (c) is due to the condition that $\phi$ is $c$-Lipschitz. Finally, by the argument in Lemma 3.6, we know that the sum of $\upsilon_k, \gamma_k$ follows beta distribution $Beta(\frac{K}{2}, \frac{d-K}{2})$. Together we complete the proof. □

## F. Proof of Lemma 3.6

**Lemma.** *Let $A, B$ be any unit vector and $\sum_{k=1}^{K} u_k u_k^T \sim \mathsf{Unif}(V_k(\mathbb{R}^d))$. Denoting $\upsilon_k = (\langle A, u_k \rangle)^2$ and $\gamma_k = (\langle B, u_k \rangle)^2$. Then for any $\epsilon \geq 0$ and $d \geq 2K$,*

$$\mathbb{P}\left( \sqrt{1 - \sum_{k=1}^{K} \upsilon_k + c^2 \sum_{k=1}^{K} \gamma_k} \leq \sqrt{1 - (1-c^2)\frac{K}{d} + \epsilon(1+c^2)\frac{K}{d}} \right) \tag{52}$$

$$\geq 1 - 2\exp(-\frac{3\epsilon^2 dK}{12(d-K) + 8\epsilon(d-2K)}). \tag{53}$$

*Proof.* First observe that $\sum_{k=1}^{K} u_k u_k^T \stackrel{d}{=} \mathbf{O} \sum_{k=1}^{K} e_k e_k^T$, where $\mathbf{O} \sim \mathsf{Unif}(V_d(\mathbb{R}^d))$ follows the uniform distribution over all possible rotation matrices in $\mathbb{R}^d$. Then for any unit vector $A$, we have

$$[\langle A, u_1 \rangle, \ldots, \langle A, u_1 \rangle] \stackrel{d}{=} [\langle A, Oe_1 \rangle, \ldots, \langle A, Oe_K \rangle] = [\langle O^T A, e_1 \rangle, \ldots, \langle O^T A, e_K \rangle]. \tag{54}$$

Note that $O^T A$ follows the distribution of $\mathsf{Unif}(\mathbb{S}^{d-1})$, which implies that we have the following characterization.

$$O^T A \stackrel{d}{=} \frac{Z}{\|Z\|}, \tag{55}$$

where $Z \sim N(0, I_d)$ is a standard normal vector. As a result, we have

$$\sum_{k=1}^{K} \upsilon_k = \sum_{k=1}^{K} (\langle A, u_k \rangle)^2 \stackrel{d}{=} \sum_{k=1}^{K} (\langle O^T A, e_k \rangle)^2 \stackrel{d}{=} \frac{\sum_{k=1}^{K} Z_k^2}{\|Z\|^2} \sim Beta(\frac{K}{2}, \frac{d-K}{2}), \tag{56}$$

where $Z_k = \langle Z, e_k \rangle$. The same arguement hold for $\gamma_k$, where we also have $\sum_{k=1}^{K} \gamma_k \sim Beta(\frac{K}{2}, \frac{d-K}{2})$. Next we introduce the following tail bound for the beta distribution.

**Theorem F.1** (Tail bound for beta distribution, simplification of Theorem 1 in (Skorski, 2023))**.** *Let $X \sim Beta(\alpha, \beta)$ and $\beta \geq \alpha$. Define $v = \frac{\alpha\beta}{(\alpha+\beta)^2(\alpha+\beta+1)}$ and $c = \frac{2(\beta-\alpha)}{(\alpha+\beta)(\alpha+\beta+2)}$. Then we have*

$$\mathbb{P}(X > \mathbb{E}X + \epsilon) \wedge \mathbb{P}(X < \mathbb{E}X - \epsilon) \leq \exp\left(-\frac{\epsilon^2}{2(v + \frac{c}{3}\epsilon)}\right). \tag{57}$$

Note that $\mathbb{E}X = \frac{\alpha}{\alpha+\beta}$ for $X \sim Beta(\alpha, \beta)$. As a result, $\mathbb{E} \sum_{k=1}^{K} v_k = \frac{K}{d}$. First we have

$$v = \frac{\frac{K(d-K)}{4}}{(\frac{d}{2})^2(\frac{d}{2}+1)} = \frac{2K(d-K)}{d^2(d+2)} \leq \frac{2K(d-K)}{d^3}, \quad c = \frac{d-2K}{\frac{d}{2}(\frac{d}{2}+2)} = \frac{4(d-2K)}{d(d+4)} \leq \frac{4(d-2K)}{d^2}. \tag{58}$$

By reparametrize $\epsilon \leftarrow \epsilon \frac{K}{d}$ in Theorem F.1, we can simplify the bound as follows

$$\mathbb{P}\left(\sum_{k=1}^{K} v_k > (1+\epsilon)\frac{K}{d}\right) \wedge \mathbb{P}\left(\sum_{k=1}^{K} v_k < (1-\epsilon)\frac{K}{d}\right) \leq \exp\left(-\frac{\epsilon^2 \frac{K^2}{d^2}}{2(v + \frac{cK}{3d}\epsilon)}\right) \tag{59}$$

$$\leq \exp\left(-\frac{\epsilon^2 \frac{K^2}{d^2}}{2(\frac{2K(d-K)}{d^3} + \frac{K}{3d}\frac{4(d-2K)}{d^2}\epsilon)}\right) = \exp\left(-\frac{3\epsilon^2 K d}{12(d-K) + 8(d-2K)\epsilon}\right). \tag{60}$$

Note that the same bound hold for $\gamma$ part. Finally, by leveraging the lower tail bound for $\sum_{k=1}^{K} v_k$, upper tail bound for $\sum_{k=1}^{K} \gamma_k$ and applying Boole's inequality we complete the proof. $\square$

## G. Proof of Lemma 3.7

**Lemma** (Forward Wasserstein distance tracking). *Let $w_t, w_t'$ be the process defined in* (4) *and* (6).

$$W_\infty(w_t, w_t') \leq \min(2R, \frac{2\eta\Delta t}{\sqrt{K}}) \tag{61}$$

*Proof.* We will start with the definition of Wasserstein distance. Let $\nu_t, \nu_t'$ be the distribution of $w_t, w_t'$ respectively.

$$W_\infty(w_t, w_t') = \inf_{\gamma \in \Gamma(\nu_t, \nu_t')} \operatorname*{ess\,sup}_{(w_t, w_t') \sim \gamma} \|w_t - w_t'\|. \tag{62}$$

Note that the infimum is taking over all possible coupling between two stochastic processes $w_t, w_t'$. Hence, choosing any specific coupling will lead to an upper bound. We choose a specific coupling such that all the noise $G_t$ and $G_{t,k}$ are identical to $G_t'$ and $G_{t,k}'$, as well as the $u_{t,k} = u_{t,k}'$. Let us denote such a coupling as $\gamma^\star$, then we have

$$W_\infty(w_t, w_t') \leq \operatorname*{ess\,sup}_{(w_t, w_t') \sim \gamma^\star} \|w_t - w_t'\| \tag{63}$$

$$\leq \operatorname*{ess\,sup}_{(w_t, w_t') \sim \gamma^\star} \|w_{t-1} - \frac{\eta}{K}\sum_{k=1}^{K} \hat{g}_t(w_{t-1}; u_{t,k}) - w_{t-1}' + \frac{\eta}{K}\sum_{k=1}^{K} \hat{g}_t'(w_{t-1}'; u_{t,k})\| \tag{64}$$

$$\leq \operatorname*{ess\,sup}_{(w_t, w_t') \sim \gamma^\star} \|w_{t-1} - w_{t-1}'\| + \operatorname*{ess\,sup}_{(w_t, w_t') \sim \gamma^\star} \|\frac{\eta}{K}\sum_{k=1}^{K} \hat{g}_t(w_{t-1}; u_{t,k}) - \frac{\eta}{K}\sum_{k=1}^{K} \hat{g}_t'(w_{t-1}'; u_{t,k})\|. \tag{65}$$

The first term will contribute to a recursive argument. The second term can be further upper bounded as follows:

$$\|\frac{\eta}{K}\sum_{k=1}^{K} \hat{g}_t(w_{t-1}; u_{t,k}) - \frac{\eta}{K}\sum_{k=1}^{K} \hat{g}_t'(w_{t-1}'; u_{t,k})\|^2 \tag{66}$$

$$= (\frac{\eta}{K})^2 \sum_{k=1}^{K} \|\frac{1}{n}\sum_{i=1}^{n} \text{clip}\left(\frac{\ell_i(w_{t-1} + \xi u_{t,k}) - \ell_i(w_{t-1} - \xi u_{t,k})}{2\xi}; \Delta\right) \tag{67}$$

$$- \text{clip}\left(\frac{\ell_i'(w_{t-1}' + \xi u_{t,k}) - \ell_i'(w_{t-1}' - \xi u_{t,k})}{2\xi}; \Delta\right) u_{t,k}\|^2 \tag{68}$$

$$\leq (\frac{\eta}{K})^2 \sum_{k=1}^{K} (2\Delta)^2 = \frac{\eta^2(2\Delta)^2}{K}. \tag{69}$$

Note that the second equality and last inequality are due to our choice that $u_{t,k}$ are orthonormal. As a result, we have

$$W_\infty(w_t, w'_t) \leq \underset{(w_t, w'_t) \sim \gamma^\star}{\mathrm{ess\,sup}} \|w_t - w'_t\| \tag{70}$$

$$\leq \underset{(w_t, w'_t) \sim \gamma^\star}{\mathrm{ess\,sup}} \|w_{t-1} - w'_{t-1}\| + \frac{2\eta\Delta}{\sqrt{K}} \tag{71}$$

$$\cdots \leq \underset{(w_t, w'_t) \sim \gamma^\star}{\mathrm{ess\,sup}} \|w_0 - w'_0\| + \frac{2\eta\Delta t}{\sqrt{K}} = \frac{2\eta\Delta t}{\sqrt{K}}, \tag{72}$$

where the last step is due to the fact that both processes $w_t, w'_t$ have the same initialization $w_0 = w'_0$. Together we complete the proof. $\qquad\square$

## H. Minibatch extensions

Now, we discuss the minibatch generalizations of Noisy-ZOGD (Theorem 3.2), which we called the corresponding algorithm Noisy-ZOSGD (under subsampling without replacement).

**Subsampling without replacement.** For this strategy, the Noisy-ZOSGD update is defined as follows.

$$w_{t+1} = \Pi_{\mathcal{B}_R} \left[ w_t - \frac{\eta}{K} \sum_{k=1}^K \hat{g}_t(w_t; u_{t,k}, B_t) + \frac{\eta}{\sqrt{K}} \sum_{k=1}^K G_{t,k}^{(1)} u_{t,k} + \frac{\eta}{\sqrt{d}} G_t^{(2)} \right], \tag{73}$$

$$\text{where } \hat{g}_t(w; u, B) = \frac{1}{b} \sum_{i \in B} \mathrm{clip}\left( \frac{\ell_i(w + \xi u) - \ell_i(w - \xi u)}{2\xi}; \Delta \right) u. \tag{74}$$

Here, the minibatch $B$ is of size $b$ and is sampled from the full index set $[n]$ without replacement independently for each time step. The rest setting is the same as the full batch cases, which can be viewed as the special case of $B = [n]$. Before we introduce the corresponding privacy guarantee, we first need to introduce the Sampled Gaussian Mechanism (Mironov et al., 2019).

**Definition H.1** (Rényi divergence of Sampled Gaussian Mechanism). For any $\alpha > 1$, mixing probability $q \in (0, 1)$ and noise parameter $\sigma > 0$, define

$$S_\alpha(q, \sigma) = D_\alpha(N(0, \sigma^2) \| (1 - q)N(0, \sigma^2) + qN(1, \sigma^2)). \tag{75}$$

Note that $S_\alpha$ can be computed in practice with a numerically stable procedure for precise computation (Mironov, 2017). Now we are ready to state the result for subsampling without replacement.

**Theorem H.2** (DP guarantee of Noisy-ZOSGD). *Assume $\ell(\cdot, x)$ are $M$-smooth and $\Delta$-Lipschitz for the first argument and all possible data $x$. The Noisy-ZOSGD under subsampling without replacement update (73) is $(\rho_\alpha^\star + \frac{\log(1/\delta_p)}{\alpha - 1}, \delta_p + \delta_f)$-DP for any $\alpha > 1, d \geq 2K, \vartheta \geq 0, \delta_p \in (0, 1)$, where*

$$\rho_\alpha^\star = \min_{\tau, \beta_t, a_t} \sum_{t=\tau}^{T-1} \left( KS_\alpha(\frac{b}{n}, \frac{\sqrt{K\beta_t}\sigma b}{2\Delta}) + \frac{\alpha a_t^2 d}{2\eta^2(1 - \beta_t)\sigma^2} \right), \ s.t. \ \tau \in \{0, \ldots, T-1\}, \ \beta_t \in [0, 1],$$

$$a_t, z_t \geq 0 \ \forall t \geq \tau, \ z_{t-1} = c_1^{-1}(z_t + a_t - c_2), \ z_T = 0, \ z_\tau \geq \min(2R, \frac{2\eta\Delta\tau}{\sqrt{K}}), \ c = 1 + \frac{\eta M}{K},$$

$$c_1 = \sqrt{1 - (1 - c^2)\frac{K}{d} + (1 + c^2)\frac{K}{d}}, \ c_2 = \eta M\xi,$$

$$\delta_f = 2(T - \tau^\star)\exp\left( -\frac{3^2 dK}{12(d - K) + 8(d - 2K)} \right),$$

*where $\tau^\star$ is the resulting $\tau$ of the optimization above. If $\ell$ is also convex and choose $\eta \leq 2K/M$, we have $c = 1$. If $\ell$ is also $m$-strongly convex and choose $\eta \leq K/M$, we have $c = 1 - \frac{\eta m}{K}$.*

The proof mainly combines the proof of the first-order PABI analysis (Chien & Li, 2025; Altschuler & Talwar, 2022) and the proof of our Theorem 3.2. Before we state our proof, let us first introduce a technical lemma before we introduce our proof.

Note that the original Sampled Gaussian Mechanism is defined in one dimension. (Altschuler & Talwar, 2022) extends this notion to a higher dimension by identifying the worst-case scenario therein. Alternatively, one can directly work with high-dimensional Sampled Gaussian Mechanism as in (Altschuler et al., 2024).

**Lemma H.3** (Extrema of Sampled Gaussian Mechanism, Lemma 2.11 in (Altschuler & Talwar, 2022)). *For any $\alpha > 1$, $q \in (0, 1)$ and the noise parameter $\sigma > 0$, dimension $d \in \mathbb{N}$ and radius $R > 0$,*

$$\sup_{\mu \in \mathcal{P}(B_R)} D_\alpha(N(0, \sigma^2 I_d) || (1 - q)N(0, \sigma^2 I_d) + q(N(0, \sigma^2 I_d) * \mu)) = S_\alpha(q, \sigma/R), \tag{76}$$

*where $\mathcal{P}(B_R)$ denotes set of all Borel probability distributions over the $\ell_2$ ball of radius $R$ in $\mathbb{R}^d$.*

In practice, $S_\alpha(q, \sigma)$ is computed via numerical integral for the tightest possible privacy accounting. Nevertheless, to have a better understanding of the quantity, Lemma 2.12 in (Altschuler & Talwar, 2022) shows that it is upper bounded by $2\alpha q^2/\sigma^2$ for some regime of $(\alpha, \sigma, q)$. In what follows, we keep the notation of $S_\alpha(q, \sigma)$ but is useful to keep this simplified upper bound in mind.

Now we are ready to state our proof.

*Proof.* We repeat the same proof as in those of Theorem 3.2 except for the Rényi cost of regarding $b_t, Y_t$. The second terms of $\rho_\alpha$ will be bounded the same way as in Theorem 3.2, and the only different part is the way of upper bounding the first term in $\rho_\alpha$.

For the first term, we first observe that the sensitivity $m_{t,k}$ of each direction $u_{t,k}$ is

$$m_{t,k} := \tag{77}$$
$$\left\| \frac{\eta}{bK} \left( \text{clip}\left( \frac{\ell_i'(w + \xi u_{t,k}) - \ell_i'(w - \xi u_{t,k})}{2\xi}; \Delta \right) - \text{clip}\left( \frac{\ell_i'(w + \xi u_{t,k}) - \ell_i'(w - \xi u_{t,k})}{2\xi}; \Delta \right) \right) u_{t,k} \right\|$$
$$\leq \frac{2\eta\Delta}{bK}. \tag{78}$$

As a result, we can bound the first term in $\rho_\alpha$ as follows (let $G'_{t,k}$ be $G_{t,k}$ with shift $m_{t,k}$)

$$D_\alpha(G_{\tau:T-1,1:K} || G'_{\tau:T-1,1:K}) \tag{79}$$
$$\leq \sum_{t=\tau}^{T-1} \sup_{g_{\tau:t-1,1:K}} D_\alpha(G_{t,1:K}|_{G_{\tau:t-1,1:K}=g_{\tau:t-1,1:K}} || G'_{t,1:K}|_{G'_{\tau:t-1,1:K}=g_{\tau:t-1,1:K}}), \tag{80}$$
$$\overset{(a)}{\leq} K \sum_{t=\tau}^{T-1} \sup_{g_{\tau:t-1,1}} D_\alpha(G_{t,1:K}|_{G_{\tau:t-1,1:K}=g_{\tau:t-1,1:K}} || G'_{t,1:K}|_{G'_{\tau:t-1,1}=g_{\tau:t-1,1}}), \tag{81}$$
$$= K \sum_{t=\tau}^{T-1} D_\alpha(N(0, \frac{\eta^2}{K}\beta_t\sigma^2 I_K) || (1 - \frac{b}{n})N(0, \frac{\eta^2}{K}\beta_t\sigma^2 I_K) + \frac{b}{n}N(m_{t,1}, \frac{\eta^2}{K}\beta_t\sigma^2 I_K)) \tag{82}$$
$$\leq \sum_{t=\tau}^{T-1} K S_\alpha(\frac{b}{n}, \frac{\sqrt{K\beta_t}\sigma b}{2\Delta}), \tag{83}$$

where (a) is again by composition theorem and by the i.i.d. property of the Gaussian noise $G_{t,k}$; the rest analysis we use the fact that $\|m_{t,1}\| \leq \frac{2\eta\Delta}{bK}$ almost surely. Hence, one can apply Lemma H.3 for the last inequality. Together we complete the proof. □

**Remark.** The analysis above can be slightly simplified to eliminate the effect of $K$. Thanks to our choice of orthonormal $u_{t,k}$, we can instead leverage the standard Gaussian mechanism result in the subspace spanned by $u_{t,k}$. For each time step $t$, the corresponding sensitivity is $\frac{2\eta\Delta}{b\sqrt{K}}$ (converting an $\ell_\infty$ norm to $\ell_2$ norm will loose a $\sqrt{K}$ factor for $K$-dimensional subspace), and thus, following the similar analysis as above leads to a bound

$$D_\alpha(G_{\tau:T-1,1:K}\|G'_{\tau:T-1,1:K}) \le \sum_{t=\tau}^{T-1} S_\alpha(\frac{b}{n}, \frac{\sqrt{\beta_t}\sigma b}{2\Delta}). \tag{84}$$

In practice, one can choose the smaller bound for computing the final privacy loss. Note that when $b = n$, $S_\alpha(1, \frac{\sqrt{\beta_t}\sigma b}{2\Delta}) = \frac{\alpha}{2}(\frac{2\Delta}{\sqrt{\beta_t}\sigma b})^2 = K\frac{\alpha}{2}(\frac{2\Delta}{\sqrt{K\beta_t}\sigma b})^2 = KS_\alpha(1, \frac{\sqrt{K\beta_t}\sigma b}{2\Delta})$, where the two approach coincide. For simplicity, we just choose the first approach in the theorems.

## I. Utility analysis

The analysis will mainly follow the proof of DPZero (Zhang et al., 2024a). The purpose of this section is to identify the fair tradeoff scaling between scalar and vector Gaussian noise in Noisy-ZOGD (4). We repeat the proof here merely for self-containedness. As a direct consequence, we also report the resulting privacy-utility tradeoff at the end of the section.

We start by introducing the necessary assumptions for the analysis of DPZero. 1) The loss function is $L$-Lipschitz and $M$-smooth. The averaged loss function is twice differentiable with $-H \preceq \nabla^2 L(w;\mathcal{D}) \preceq H$ for any $w \in \mathbb{R}^d$, and its minimum is finite. Here, $0 \preceq H$ is a real-valued $d$ by $d$ matrix such that $\|H\|_2 \le M$ and $Tr(H) \le r\|H\|_2$ for for some $r$ as the effective rank or the intrinsic dimension of the problem. These assumptions are exactly the Assumption 3.5 in (Zhang et al., 2024a).

For simplicity, let us denote $\bar{\ell}(x) = \frac{1}{n}\sum_{i=1}^n \ell_i(x)$ be the total loss function that we study. By $M$-smoothness of $\bar{\ell}(x)$, we know that for any unit vector $u$,

$$\frac{|\bar{\ell}(x+\xi u) - \bar{\ell}(x-\xi u)|}{2\xi} \le |\langle \nabla\bar{\ell}(x), u\rangle| + \frac{1}{2}M\xi. \tag{85}$$

Next, the Lemma C.1 in (Zhang et al., 2024a) provides a high probability bound for $|\langle \nabla\bar{\ell}(x), u\rangle|$, where $u$ is a unit vector uniformly distributed on $\mathbb{S}^{d-1}$. Note that one can also use bound for beta distribution as in Lemma 3.6, but we choose to align as much as possible to DPZero for simplicity here. The results show that for any $C_0 \ge 0$ we have

$$\mathbb{P}\left(|\langle u, \nabla\bar{\ell}(x)\rangle| \ge C_0\right) \le 2\sqrt{2\pi}\exp(-\frac{dC_0^2}{8\|\nabla\bar{\ell}(x)\|^2}). \tag{86}$$

As a result, by choosing the clipping $\Delta = C_0 + \frac{1}{2}M\xi$ we can safely remove the clipping operator with high probability after applying Boole's inequality over $nTK$ terms. Let us denote $Q_t$ the event that clipping does not happen at iteration $t$ and $Q$ the event that clipping does not happen at all $T$ iteration. This will be dealt in the end.

Now let us continue with the analysis. By Taylor's theorem and the low-rank assumption that $\forall x, -H \preceq \nabla^2\bar{\ell}(x) \preceq H$ for some positive semi-definite matrix with $Tr(H) \le rM$ and $\|H\|_2 \le M$, we have

$$\bar{\ell}(w_{t+1}) \le \bar{\ell}(w_t) + \langle \nabla\bar{\ell}(x_t), w_{t+1} - w_t\rangle + \frac{1}{2}(w_{t+1} - w_t)^T H(w_{t+1} - w_t). \tag{87}$$

By taking expectation across all randomness before iteration $t$ (including the random direction $u$ and the gaussian noise $G$), we have

$$\mathbb{E}_{\le t}[\bar{\ell}(w_{t+1})|Q_t] \le \mathbb{E}_{\le(t-1)}[\bar{\ell}(w_t)|Q_t] - \eta\mathbb{E}_{\le t}[\nabla\bar{\ell}(w_t)^T\hat{g}_t(w_t)|Q_t] + \frac{\eta^2}{2}\mathbb{E}_{\le t}[\hat{g}_t(w_t)^T H\hat{g}_t(w_t)|Q_t] \tag{88}$$

$$+ \frac{\sigma_{1,t}^2}{2}\sum_{k=1}^K \mathbb{E}_{\le t}[u_{t,k}^T H u_{t,k}|Q_t] + \frac{\sigma_{2,t}^2}{2}Tr(H). \tag{89}$$

Note that all the cross terms are 0 as $G_{t,k}, G_t$ are zero mean Gaussian independent of all the other randomness. Now, we analyze each term separately. Let us analyze the quadratic terms first.

$$\mathbb{E}_{\leq t}[\hat{g}_t(w_t)^T H \hat{g}_t(w_t)|Q_t] = \frac{1}{K} \sum_{k=1}^{K} \mathbb{E}_{\leq t}\left[\left(\frac{\bar{\ell}(w_t + \xi u_{t,k}) - \bar{\ell}(w_t - \xi u_{t,k})}{2\xi}\right)^2 u_{t,k}^T H u_{t,k}|Q_t\right] \tag{90}$$

This is due to the fact that $u_{t,k} \perp u_{t,k'}$ by definition. Then since $0 \preccurlyeq H$, by the law of total probability we have

$$\frac{1}{K} \sum_{k=1}^{K} \mathbb{E}_{\leq t}\left[\left(\frac{\bar{\ell}(w_t + \xi u_{t,k}) - \bar{\ell}(w_t - \xi u_{t,k})}{2\xi}\right)^2 u_{t,k}^T H u_{t,k}\right] \tag{91}$$

$$\geq \frac{1}{K} \sum_{k=1}^{K} \mathbb{E}_{\leq t}\left[\left(\frac{\bar{\ell}(w_t + \xi u_{t,k}) - \bar{\ell}(w_t - \xi u_{t,k})}{2\xi}\right)^2 u_{t,k}^T H u_{t,k}|Q_t\right]\mathbb{P}(Q_t). \tag{92}$$

Therefore we have

$$\mathbb{E}_{\leq t}[\hat{g}_t(w_t)^T H \hat{g}_t(w_t)|Q_t] \leq \frac{1}{\mathbb{P}(Q_t)K} \sum_{k=1}^{K} \mathbb{E}_{\leq t}\left[\left(\frac{\bar{\ell}(w_t + \xi u_{t,k}) - \bar{\ell}(w_t - \xi u_{t,k})}{2\xi}\right)^2 u_{t,k}^T H u_{t,k}\right] \tag{93}$$

$$\overset{(a)}{\leq} \frac{1}{\mathbb{P}(Q_t)K} \sum_{k=1}^{K} \mathbb{E}_{\leq t}\left[\left(2(\langle \nabla\bar{\ell}(w_t), u_{t,k}\rangle)^2 + \frac{1}{2}M^2\xi^2\right) u_{t,k}^T H u_{t,k}\right] \tag{94}$$

$$\overset{(b)}{\leq} \frac{1}{\mathbb{P}(Q_t)K} \sum_{k=1}^{K} \left(\mathbb{E}_{\leq t}[(2(\langle \nabla\bar{\ell}(w_t), u_{t,k}\rangle)^2) u_{t,k}^T H u_{t,k}] + \frac{1}{2d}M^2\xi^2 Tr(H)\right) \tag{95}$$

$$\overset{(c)}{\leq} \frac{1}{\mathbb{P}(Q_t)K} \sum_{k=1}^{K} \left(\frac{2d}{d^2(d+2)}\mathbb{E}_{\leq t}(2\nabla\bar{\ell}(w_t)^T H \nabla\bar{\ell}(w_t) + \|\nabla\bar{\ell}(w_t)\|^2 Tr(H)) + \frac{1}{2d}M^2\xi^2 Tr(H)\right) \tag{96}$$

$$\leq \frac{1}{\mathbb{P}(Q_t)K} \sum_{k=1}^{K} \left(\frac{2(2+r)M}{d(d+2)}\mathbb{E}_{\leq t}[\|\nabla\bar{\ell}(w_t)\|^2] + \frac{1}{2d}M^3\xi^2 r\right) \tag{97}$$

$$= \frac{2(2+r)M}{d(d+2)\mathbb{P}(Q_t)}\mathbb{E}_{\leq t}[\|\nabla\bar{\ell}(w_t)\|^2] + \frac{M^3\xi^2 r}{2d\mathbb{P}(Q_t)}. \tag{98}$$

where (a) is due to $M$-smoothness and the elementary inequality $(a+b)^2 \leq 2a^2 + 2b^2$. (b) and (c) are due to Lemma C.1 in (Zhang et al., 2024a).

Similarly we have

$$\sum_{k=1}^{K} \mathbb{E}_{\leq t}[u_{t,k}^T H u_{t,k}|Q_t] \leq \frac{rMK}{d\mathbb{P}(Q_t)}. \tag{99}$$

For the inner-product term, let us denote $u' = \sqrt{d}u$ and $\xi' = \frac{\xi}{\sqrt{d}}$.

$$\mathbb{E}_{\leq t}[\nabla\bar{\ell}(w_t)^T \hat{g}_t(w_t)|Q_t] = \frac{1}{K\sqrt{d}} \sum_{k=1}^{K} \mathbb{E}_{\leq t}[\nabla\bar{\ell}(w_t)^T \frac{\bar{\ell}(w_t + \xi u_{t,k}) - \bar{\ell}(w_t - \xi u_{t,k})}{2\xi} u'_{t,k}|Q_t] \tag{100}$$

$$= \frac{1}{Kd} \sum_{k=1}^{K} \mathbb{E}_{\leq t}[\nabla\bar{\ell}(w_t)^T \frac{\bar{\ell}(w_t + \xi' u'_{t,k}) - \bar{\ell}(w_t - \xi' u'_{t,k})}{2\xi'} u'_{t,k}|Q_t]. \tag{101}$$

Following the similar analysis as in (Zhang et al., 2024a), we have

$$\mathbb{E}_{\leq t}[\nabla \bar{\ell}(w_t)^T \frac{\bar{\ell}(w_t + \xi' u'_{t,k}) - \bar{\ell}(w_t - \xi' u'_{t,k})}{2\xi'} u'_{t,k} | Q_t] \tag{102}$$

$$\geq \frac{\mathbb{E}_{\leq t}[\|\nabla \bar{\ell}(w_t)\|^2]}{2\mathbb{P}(Q_t)} - \frac{M^2 \xi^2 d^3}{8\mathbb{P}(Q_t)} - \frac{\mathbb{E}_{\leq t}[\nabla \bar{\ell}(w_t)^T \frac{\bar{\ell}(w_t + \xi' u'_{t,k}) - \bar{\ell}(w_t - \xi' u'_{t,k})}{2\xi'} u'_{t,k} | \bar{Q}_t] \mathbb{P}(\bar{Q}_t)}{\mathbb{P}(Q_t)} \tag{103}$$

$$\geq \frac{\mathbb{E}_{\leq t}[\|\nabla \bar{\ell}(w_t)\|^2]}{2\mathbb{P}(Q_t)} - \frac{M^2 \xi^2 d^3}{8\mathbb{P}(Q_t)} - \frac{L^2 d \mathbb{P}(\bar{Q}_t)}{\mathbb{P}(Q_t)}, \tag{104}$$

where the last inequality is due to the following analysis. Note that by Cauchy-Schwartz inequality and the fact that $\bar{\ell}$ is $L$-Lipschitz, we have

$$\nabla \bar{\ell}(w_t)^T \frac{\bar{\ell}(w_t + \xi' u'_{t,k}) - \bar{\ell}(w_t - \xi' u'_{t,k})}{2\xi'} u'_{t,k} \leq L^2 \|u'_{t,k}\|^2 = L^2 d. \tag{105}$$

Combining all we have so far leads to

$$\mathbb{E}_{\leq t}[\bar{\ell}(w_{t+1})|Q_t] \leq \mathbb{E}_{\leq(t-1)}[\bar{\ell}(w_t)|Q_t] - \frac{\eta}{d}\left(\frac{\mathbb{E}_{\leq t}[\|\nabla \bar{\ell}(w_t)\|^2]}{2\mathbb{P}(Q_t)} - \frac{M^2 \xi^2 d^3}{8\mathbb{P}(Q_t)} - \frac{L^2 d \mathbb{P}(\bar{Q}_t)}{\mathbb{P}(Q_t)}\right) \tag{106}$$

$$+ \frac{\eta^2}{2}\left(\frac{2(2+r)M}{d(d+2)\mathbb{P}(Q_t)}\mathbb{E}_{\leq t}[\|\nabla \bar{\ell}(w_t)\|^2] + \frac{M^3 \xi^2 r}{2d\mathbb{P}(Q_t)}\right) + \frac{\sigma_{1,t}^2 rMK}{2d\mathbb{P}(Q_t)} + \frac{\sigma_{2,t}^2 rM}{2} \tag{107}$$

$$\leq \mathbb{E}_{\leq(t-1)}[\bar{\ell}(w_t)|Q_t] - \frac{\eta}{2d}(1 - \frac{\eta 2(2+r)M}{d})\frac{\mathbb{E}_{\leq t}[\|\nabla \bar{\ell}(w_t)\|^2]}{\mathbb{P}(Q_t)} + \frac{\eta M^2 \xi^2 d^2}{8\mathbb{P}(Q_t)} + \frac{\eta L^2 \mathbb{P}(\bar{Q}_t)}{\mathbb{P}(Q_t)} \tag{108}$$

$$+ \frac{\eta^2 M^3 \xi^2 r}{4d\mathbb{P}(Q_t)} + \frac{\sigma_{1,t}^2 rMK}{2d\mathbb{P}(Q_t)} + \frac{\sigma_{2,t}^2 rM}{2} \tag{109}$$

Now we choose $\eta = \frac{K}{M} \leq \frac{d}{4(2+r)M}$, where the inequality hold if $K \leq \frac{d}{4(2+r)}$. This results in $(1 - \frac{\eta 2(2+r)M}{d}) \geq 1/2$ and $\frac{2\eta Mr}{d^2} < 1$. Then we have

$$\mathbb{E}_{\leq t}[\|\nabla \bar{\ell}(w_t)\|^2] \leq \frac{4d\mathbb{P}(Q_t)}{\eta}\left(\mathbb{E}_{\leq t}[\bar{\ell}(w_t) - \bar{\ell}(w_{t+1})|Q_t]\right) + M^2 \xi^2 d^3 + 4L^2 d\mathbb{P}(\bar{Q}_t) \tag{110}$$

$$+ \frac{2drM}{\eta}\left(\sigma_{1,t}^2 \frac{K}{d} + \sigma_{2,t}^2 \mathbb{P}(Q_t)\right). \tag{111}$$

If we further assume that $|\bar{\ell}(w)| \leq B$ for any $w$. Then following the same analysis of ([Zhang et al., 2024a](#)) we have

$$\mathbb{E}_{\leq t}[\bar{\ell}(w_t) - \bar{\ell}(w_{t+1})|Q_t]\mathbb{P}(Q_t) \leq \mathbb{E}_{\leq t}[\bar{\ell}(w_t) - \bar{\ell}(w_{t+1})|Q]\mathbb{P}(Q) + 2B\mathbb{P}(\bar{Q}). \tag{112}$$

As a result we have

$$\mathbb{E}_{\leq t}[\|\nabla \bar{\ell}(w_t)\|^2] \leq \frac{4d\mathbb{P}(Q)}{\eta}\left(\mathbb{E}_{\leq t}[\bar{\ell}(w_t) - \bar{\ell}(w_{t+1})|Q]\right) + M^2 \xi^2 d^3 + (4L^2 d + \frac{8Bd}{\eta})\mathbb{P}(\bar{Q}) \tag{113}$$

$$+ \frac{2drM}{\eta}\left(\sigma_{1,t}^2 \frac{K}{d} + \sigma_{2,t}^2\right). \tag{114}$$

By averaging over $T$ iteration, we have

$$\mathbb{E}[\|\nabla \bar{\ell}(w_\tau)\|^2] = \frac{1}{T}\sum_{t=0}^{T-1}\mathbb{E}_{\leq t}[\|\nabla \bar{\ell}(w_t)\|^2] \tag{115}$$

$$\leq \frac{4d\mathbb{P}(Q)}{T\eta}\left(\mathbb{E}_{\leq t}[\bar{\ell}(w_0) - \bar{\ell}(w_T)|Q]\right) + M^2 \xi^2 d^3 + (4L^2 d + \frac{8Bd}{\eta})\mathbb{P}(\bar{Q}) + \frac{2drM}{\eta}\left(\sigma_{1,t}^2 \frac{K}{d} + \sigma_{2,t}^2\right), \tag{116}$$

where

$$\mathbb{P}(\bar{Q}) \leq 2\sqrt{2\pi}nKT \exp(-\frac{dC_0^2}{8\|\nabla\bar{\ell}(x)\|^2}). \tag{117}$$

If we plug in our parametrization of $\sigma_1^2 = \frac{\eta^2}{K}\beta\sigma^2, \sigma_2^2 = \frac{\eta^2}{d}(1-\beta)\sigma^2$ for any $\beta \in [0,1]$ as in the privacy analysis, we further have

$$\mathbb{E}[\|\nabla\bar{\ell}(w_\tau)\|^2] = \frac{1}{T}\sum_{t=0}^{T-1}\mathbb{E}_{\leq t}[\|\nabla\bar{\ell}(w_t)\|^2] \tag{118}$$

$$\leq \frac{4d\mathbb{P}(Q)}{T\eta}\left(\mathbb{E}_{\leq t}[\bar{\ell}(w_0) - \bar{\ell}(w_T)|Q]\right) + M^2\xi^2 d^3 + (4L^2 d + \frac{8Bd}{\eta})\mathbb{P}(\bar{Q}) + 2\eta r M\sigma^2, \tag{119}$$

Apparently, this bound remains unchanged for any $\beta \in [0,1]$ and $K \geq 1$, and this bound exactly matches the bound derived in (Zhang et al., 2024a) for DPZero with the scalar noise case. Thus our parametrization $\sigma_1^2 = \frac{\eta^2}{K}\beta\sigma^2, \sigma_2^2 = \frac{\eta^2}{d}(1-\beta)\sigma^2$ indeed provides the same utility bound.

### I.0.1. THE PRIVACY-UTILITY TRADE-OFF IN CLOSE-FORM

We may plug in the value of $\sigma^2$ according to our privacy bound to achieve $(\varepsilon, \delta)$-DP guarantee, which leads to

$$\frac{4d\mathbb{P}(Q)}{T\eta}\left(\mathbb{E}_{\leq t}[\bar{\ell}(w_0) - \bar{\ell}(w_T)|Q]\right) + (4L^2 d + \frac{8Bd}{\eta})\mathbb{P}(\bar{Q}) + 2\eta r M\sigma^2 \tag{120}$$

$$= \frac{4d\mathbb{P}(Q)}{T\eta}\left(\mathbb{E}_{\leq t}[\bar{\ell}(w_0) - \bar{\ell}(w_T)|Q]\right) + (4L^2 d + \frac{8Bd}{\eta})\mathbb{P}(\bar{Q}) + \frac{64rMR\sqrt{d}\Delta\log(2/\delta)}{n\varepsilon^2}. \tag{121}$$

Next we plug in the expression of $\mathbb{P}(\bar{Q})$, which gives the following upper bound

$$\frac{4d\mathbb{P}(Q)}{T\eta}\left(\mathbb{E}_{\leq t}[\bar{\ell}(w_0) - \bar{\ell}(w_T)|Q]\right) + (4L^2 d + \frac{8Bd}{\eta})2\sqrt{2\pi}nKT\exp(-\frac{d\Delta^2}{8\|\nabla\bar{\ell}(x)\|^2}) \tag{122}$$

$$+ \frac{64rMR\sqrt{d}\Delta\log(2/\delta)}{n\varepsilon^2}. \tag{123}$$

Recall that under the case $\xi \to 0$, we have $C_0 = \Delta$, the zeroth order clipping value. Now we can optimize the bound with respect to $\Delta$, where the tightest analysis is related to W Lambert function and does not have close-form. We instead choose a sub-optimal $\Delta = \frac{1}{\sqrt{d}}\sqrt{\log((4L^2 d + \frac{8Bd}{\eta})2\sqrt{2\pi}nKT)}$, where we recall that $\|\nabla\bar{\ell}(x)\| \leq L$ due to the $L$-Lipschitz assumption made by (Zhang et al., 2024a). This choice makes the second term above at scale $O(1)$. So, the dominant term will be the last one, which is

$$\frac{64rMR\sqrt{d}\log(2/\delta)}{n\varepsilon^2}\frac{1}{\sqrt{d}}\sqrt{\log((4L^2 d + \frac{8Bd}{\eta})2\sqrt{2\pi}nKT)} = O(\frac{rR\log(2/\delta)}{n\varepsilon^2}\sqrt{\log(\frac{dnKT}{\eta})}). \tag{124}$$

Finally, note that we can choose $\eta = \frac{K}{M}$ whenever $K \leq \frac{d}{8(2+r)}$. On the other hand, we also need $K = \Omega(\log(T/\delta))$. This implies that choosing $T = O(d)$ is valid. Plug-in the choice $K = \Theta(\log(T/\delta)) = \Theta(\log(d/\delta))$ and $T = \Theta(d)$, we now have

$$\frac{64rMR\sqrt{d}\log(2/\delta)}{n\varepsilon^2}\frac{1}{\sqrt{d}}\sqrt{\log((4L^2d+\frac{8Bd}{\eta})2\sqrt{2\pi}nKT)} = O(\frac{rR\log(2/\delta)}{n\varepsilon^2}\sqrt{\log(d^2n)}). \tag{125}$$

Recall that now the first term will be

$$\frac{4d\mathbb{P}(Q)}{T\eta}\left(\mathbb{E}_{\leq t}[\bar{\ell}(w_0)-\bar{\ell}(w_T)|Q]\right) = O(\frac{1}{\log(d/\delta)}\left(\bar{\ell}(w_0)-\bar{\ell}(w^\star)\right)). \tag{126}$$

We can see that the third term is the dominant one, where it is of scale

$$O\left(\frac{rR\log(2/\delta)}{n\varepsilon^2}\sqrt{\log(nd)}\right) = O(r\sqrt{\log(d)}), \tag{127}$$

where the RHS only cares about the asymptotic of $r$ and $d$. This bound is better than DPZero's bound $O\left(\log(d)\sqrt{r}\right)$ whenever $r = o(\sqrt{\log(d)})$. Also note that we will use roughly $K = O(\log(d/\delta))$ zeroth-order gradients per iteration. It is possible to derive a better dependency with respect to $r, d$ if one jointly optimizes the choice of $K, T$ for all three terms simultaneously. We leave such an effort as an interesting future work.

## J. Comparison with PABI bound for Noisy-GD

Here, we provide a rough comparison of the privacy bound obtained by our paper for Noisy-ZOGD and the privacy bound for Noisy-GD in (Altschuler & Talwar, 2022). Let us first recap on the Noisy-ZOGD and Noisy-GD updates.

$$\text{Noisy-ZOGD: } w_{t+1} = \Pi_{\mathcal{B}_R}\left[w_t - \frac{\eta}{K}\sum_{k=1}^{K}\hat{g}_t(w_t; u_{t,k})\right.$$

$$\left. + \frac{\eta}{\sqrt{K}}\sum_{k=1}^{K}G_{t,k}^{(1)}u_{t,k} + \frac{\eta}{\sqrt{d}}G_t^{(2)}\right],$$

$$\text{Noisy-GD: } w_{t+1} = \Pi_{\mathcal{B}_R}\left[w_t - \eta g_t + \eta G_t\right],$$

where $\mathcal{B}_R$ is the $\ell_2$ ball of radius $R$ centered at the origin, $G_{t,k}^{(1)} \sim \mathcal{N}(0, \beta_t\sigma^2)$, $G_t^{(2)} \sim \mathcal{N}(0, (1-\beta_t)\sigma^2 I_d)$, and $G_t \sim \mathcal{N}(0, \sigma^2 I_d)$. Note that $g_t$ is the gradient evaluation at $w_t$. The vectors $\{u_{t,k}\}_{k=1}^{K}$ are orthonormal and drawn uniformly from the Stiefel manifold $V_K(\mathbb{R}^d)$; when $K = 1$, this reduces to the uniform distribution on the sphere, i.e., $u_{t,k} \sim \text{Unif}(\mathbb{S}^{d-1})$.

First, note that our noise scaling for Noisy-ZOGD and Noisy-GD stated in (Altschuler & Talwar, 2022) is different. To make a fair comparison, we know that Noisy-ZOGD should match Noisy-GD for $\xi \to 0$, $K = d$, and $\beta_t = 0$. In this case, $\sum_{k=1}^{d}\hat{g}_t(w_t; u_{t,k}) = g_t$, which is a simple change of basis. As a result, we can see that the noise variance $\sigma^2$ in Noisy-ZOGD should be replaced by $\frac{\sigma^2}{\sqrt{d}}$ for a fair comparison. In this setting, our Corollary 3.3 gives the following privacy bound:

$$O\left(\sqrt{\frac{\Delta^2\log(1/\delta)}{n^2\sigma^2}}\min(\sqrt{d}T, \frac{MRnd}{K\Delta})\right) \tag{128}$$

In the meantime, Remark 1.4 of (Altschuler & Talwar, 2022) gives

$$O\left(\sqrt{\frac{\Delta^2\log(1/\delta)}{n^2\sigma^2}}\min(T, \frac{MRn}{\Delta})\right) \tag{129}$$

There are several remarks. First, note that it is still hard to make a direct comparison for these two bounds, as we have not calibrated the noise rigorously based on some utility bound analysis. Second, the step size used in Noisy-ZOGD is $\eta = K/M$ while Noisy-GD use $1/M$. Third, the $\ell_2$ sensitivity for $\widehat{g}_t$ is $\sqrt{K}$ times $\Delta$, which is different from the standard clipping in Noisy-GD. Nevertheless, we can see that for $T \to \infty$, the privacy bound for Noisy-ZOGD at best matches the one provided by Noisy-GD, where it happens when $K = d$.

## K. Discussion on i.i.d. updates

In this section, we discuss and analyze the scenario that the update directions $\{u_{t,k}\}_{t\in[T],k\in[K]}$ in (4) are chosen i.i.d. from $\mathrm{Unif}\left(\mathbb{S}^d\right)$. We prove concentration lemmas similar but looser to Lemma 3.5 and Lemma 3.6, which can then be plugged in the proof of Theorem 3.2 and obtain a convergent Rényi DP bound. We begin with a few concentration bounds that would be useful in the proof.

**Definition K.1** (Jin et al. (2019))**.** A random vector $u \in \mathbb{R}^d$ is $\sigma$-*norm-SubGaussian*, if

$$\mathbb{P}\left(\|u - \mathbb{E}\left[u\right]\| \geq t\right) \leq 2e^{-\frac{t^2}{2\sigma^2}}.$$

**Lemma K.2.** *[Corollary 7 of Jin et al. (2019)] Let $X_1, X_2, ..., X_K$ be i.i.d. random vectors in $\mathbb{R}^d$ satisfying $\sigma$-norm-subGaussian. Then with probability at least $1 - \delta$, it holds that*

$$\left\|\sum_{k=1}^{K} X_k\right\| \leq \sqrt{K\sigma^2 \log\left(\frac{2d}{\delta}\right)}. \tag{130}$$

Note that Lemma K.2 is slightly modified from Corollary 7 of Jin et al. (2019), where we specify the absolute constant (which is 1) by following the analysis of Lemma 5.5 of (Vershynin, 2010) in order to facilitate our non-asymptotic bounds.

**Lemma K.3.** *Let $B_k \overset{i.i.d.}{\sim} \mathrm{Beta}\left(\frac{1}{2}, \frac{d-1}{2}\right)$. Then it holds that*

$$\mathbb{P}\left\{\sum_{k=1}^{K} B_k \geq \left(0.15 - \frac{\log(1/\delta)}{K}\right)\right)_+ \cdot \frac{K}{20\left(d + 1 + \sqrt{d-2}\right)}\right\} \geq 1 - \delta,$$

*where $(\cdot)_+$ denotes $\max(\cdot, 0)$.*

*Proof.* Let $B \sim \mathrm{Beta}\left(\frac{1}{2}, \frac{d-1}{2}\right)$. Then $B \overset{\text{(d)}}{=} \frac{W_1}{W_1 + W_2}$ where $W_1 \sim \chi^2\left(1\right)$ and $W_2 \sim \chi^2\left(d - 1\right)$ are independent. Since $W_1 \overset{\text{(d)}}{=} Z^2$ for $Z \sim \mathcal{N}(0, 1)$, by the lower bound of the Q-function, it holds that

$$\mathbb{P}\left\{W_1 \geq \kappa_1\right\} = \mathbb{P}\left\{|Z| \geq \sqrt{\kappa_1}\right\} = 2 \cdot Q\left(\sqrt{\kappa_1}\right) \geq \left(\frac{\sqrt{\kappa_1}}{1 + \kappa_1}\right)\frac{e^{-\kappa_1/2}}{\sqrt{2\pi}}.$$

On the other hand, according to the quantile upper bound on the $\chi^2$ random variables (see, for instance, Theorem A of (Inglot, 2010)), we have

$$\mathbb{P}\left\{W_2 \leq (d - 2) + 2\log\left(1/\beta\right) + 2\sqrt{(d-2)\log\left(1/\beta\right)}\right\} \geq 1 - \beta.$$

These together imply

$$\mathbb{P}\left\{B \geq \frac{\kappa_1}{\kappa_1 + (d-2) + 2\log\left(1/\beta\right) + 2\sqrt{(d-2)\log\left(1/\beta\right)}}\right\} \tag{131}$$

$$\geq \mathbb{P}\left\{\frac{W_1}{W_1 + W_2} \geq \frac{\kappa_1}{\kappa_1 + (d-2) + 2\log\left(1/\beta\right) + 2\sqrt{(d-2)\log\left(1/\beta\right)}}\right\} \tag{132}$$

$$\geq (1 - \beta) \cdot \left(\frac{\sqrt{\kappa_1}}{1 + \kappa_1}\right)\frac{e^{-\kappa_1/2}}{\sqrt{2\pi}} \triangleq P_e. \tag{133}$$

Next, write $G_k \triangleq \mathbb{1}_{\left\{ B_k \geq \frac{\kappa_1}{\kappa_1 + (d-2) + 2\log(1/\beta) + 2\sqrt{(d-2)\log(1/\beta)}} \right\}}$ so that $\mathbb{E}[G_k] \geq P_e$ and $G_k \in \{0, 1\}$ (i.e., $G_k \overset{\text{i.i.d.}}{\sim}$ Ber $(\mathbb{E}[G_k])$). Then we have

$$\mathbb{P}\left\{ \sum_{k=1}^{K} B_k \geq \tau \cdot \frac{\kappa_1}{\kappa_1 + (d-2) + 2\log(1/\beta) + 2\sqrt{(d-2)\log(1/\beta)}} \right\}$$

$$\geq 1 - \mathbb{P}\left\{ \sum_k G_k < \tau \right\}$$

$$\geq 1 - e^{-2K(P_e - \tau/K)^2},$$

where the last inequality is due to the following Hoeffding's inequality:

$$\mathbb{P}\left\{ \sum_k G_k \leq \tau \right\} \leq e^{-2K(\mathbb{E}[G_1] - \tau/K)^2}.$$

Setting $\tau = K \cdot \left( P_e - \frac{\log(1/\delta)}{2K} \right)_+$, $\beta = 1/e$, and $\kappa_1 = 1$ yields

$$\mathbb{P}\left\{ \sum_{k=1}^{K} B_k \geq \left( 0.15 - \frac{\log(1/\delta)}{K} \right)_+ \cdot \frac{K}{20\left(d + 1 + \sqrt{d-2}\right)} \right\} \geq 1 - \delta.$$

$\square$

**Lemma K.4.** *Let $v \in \mathbb{R}^d$ and $u_1, u_2, ..., u_K \overset{\text{i.i.d.}}{\sim}$ Unif $\left(\mathbb{S}^{d-1}\right)$. Then, it holds that, with probability at least $1 - \beta$,*

$$\left\| v - \sum_{k=1}^{K} \langle v, u_k \rangle u_k \right\|_2^2 \tag{134}$$

$$\leq \|v\|_2^2 \left( \frac{K}{2d+6} \log\left(\frac{4d}{\delta}\right) + \left( 1 - \left( 0.15 - \frac{\log(2/\delta)}{K} \right)_+ \cdot \frac{K}{20\left(d + 2 + \sqrt{d-1}\right)} \right)^2 \right). \tag{135}$$

*Proof.* By the rotational invariance of $u_k$, without loss of generality, we assume $v = R \cdot e_1$ for some $R > 0$ and let

$$u_k = \frac{1}{\sqrt{\left(z_1^k\right)^2 + \cdots + \left(z_d^{(k)}\right)^2}} \begin{bmatrix} z_1^{(k)} \\ z_2^{(k)} \\ \vdots \\ z_d^{(k)} \end{bmatrix},$$

for $k = 1, ..., K$.

Then

$$\langle v, u_k \rangle u_k = \frac{R}{\left(z_1^{(k)}\right)^2 + \cdots + \left(z_d^{(k)}\right)^2} \begin{bmatrix} \left(z_1^{(k)}\right)^2 \\ z_1^{(k)} \cdot z_2^{(k)} \\ \vdots \\ z_1^{(k)} \cdot z_d^{(k)} \end{bmatrix} \triangleq w_k + w_k^\perp,$$

where

$$w_k \triangleq \frac{R}{\left(z_1^{(k)}\right)^2 + \cdots + \left(z_d^{(k)}\right)^2} \begin{bmatrix} \left(z_1^{(k)}\right)^2 \\ 0 \\ \vdots \\ 0 \end{bmatrix}$$

and

$$w_k^\perp \triangleq \frac{R}{\left(z_1^{(k)}\right)^2 + \cdots + \left(z_d^{(k)}\right)^2} \begin{bmatrix} 0 \\ z_1^{(k)} \cdot z_2^{(k)} \\ \vdots \\ z_1^{(k)} \cdot z_d^{(k)} \end{bmatrix}.$$

Since $\langle w_k, w_{k'}^\perp \rangle = 0$ and $\langle v, w_k^\perp \rangle = 0$ for any $k, k' \in [K]$, we can write

$$\left\| v - \frac{1}{K} \sum_{k=1}^K \langle v, u_k \rangle u_k \right\|_2^2 = \underbrace{\left\| v - \sum_{k=1}^K w_k \right\|_2^2}_{(a)} + \underbrace{\left\| \sum_{k=1}^K w_k^\perp \right\|_2^2}_{(b)}. \tag{136}$$

Next, we bound each term separately.

**Bounding (a).** Observe that $(a)$ can be written as

$$(a) = R^2 \left( 1 - \sum_{k=1}^K \frac{\left(z_1^{(k)}\right)^2}{\left(z_1^{(k)}\right)^2 + \cdots + \left(z_d^{(k)}\right)^2} \right)^2 \tag{137}$$

$$\triangleq R^2 \left( 1 - \sum_{k=1}^K B_k \right)^2, \tag{138}$$

where

$$B_k \triangleq \frac{\left(z_1^{(k)}\right)^2}{\left(z_1^{(k)}\right)^2 + \cdots + \left(z_d^{(k)}\right)^2}$$

follows a Beta $\left(\frac{1}{2}, \frac{d-1}{2}\right)$ distribution. Applying Lemma K.3, we conclude that, with probability at least $1 - \delta_a$

$$(a) = \left\| v - \sum_{k=1}^K w_k \right\|_2^2$$

$$= \|v\|^2 \left( 1 - \sum_k B_k \right)^2$$

$$\leq \|v\|_2^2 \cdot \left( 1 - \left( 0.15 - \frac{\log(1/\delta_a)}{K} \right) \right)_+ \frac{K}{20 \left( d + 2 + \sqrt{d-1} \right)} \right)^2,$$

where $(\cdot)_+$ denotes $\max(\cdot, 0)$.

**Bounding (b).** To bound (b), we first show that $w_k^\perp$ is norm-sub-Gaussian and then apply Hoeffding-type inequalities.

Let

$$w^\perp \triangleq \frac{R}{\sum_{j=1}^d z_j^2} \begin{bmatrix} 0 \\ z_1 z_2 \\ \vdots \\ z_1 z_d \end{bmatrix},$$

where $z_j \stackrel{\text{i.i.d.}}{\sim} \mathcal{N}(0,1)$ for $j \in [d]$. Then

$$\|w^\perp\|_2^2 = R^2 \frac{z_1^2 \left( \sum_{j=2}^d z_j^2 \right)}{\left( \sum_{j=1}^d z_j^2 \right)^2} \leq R^2 \frac{z_1^2 \left( \sum_{j=1}^d z_j^2 \right)}{\left( \sum_{j=1}^d z_j^2 \right)^2} = r^2 \frac{z_1^2}{\sum_{j=1}^d z_j^2} \triangleq B,$$

where $B \sim \text{Beta}\left(\frac{1}{2}, \frac{d-1}{2}\right)$. As a result, by Marchal & Arbel (2017, Theorem 1), $w^\perp$ is $\frac{1}{4(1/2+d/2+1)} = \frac{1}{2d+6}$ norm-sub-Gaussian, so applying Lemma K.2 yields

$$\left\| \sum_{k=1}^K w_k^\perp \right\|_2^2 \leq \|v\|^2 \frac{K}{2d+6} \log\left(\frac{2d}{\delta_b}\right),$$

with probability at least $1 - \delta_b$.

Combining the upper bound on (a) and (b), we obtain that with probability at least $1 - \delta$,

$$\left\| v - \sum_{k=1}^K \langle v, u_k \rangle u_k \right\|_2^2 \tag{139}$$

$$\leq \|v\|_2^2 \left( \frac{K}{2d+6} \log\left(\frac{4d}{\delta}\right) + \left( 1 - \left( 0.15 - \frac{\log(2/\delta)}{K} \right)_+ \cdot \frac{K}{20\left(d + 2 + \sqrt{d-1}\right)} \right)^2 \right). \tag{140}$$

$\square$

With the above lemma in hands, we can now prove similar bounds as in Lemma 3.5 and Lemma 3.6. Following the proof of Lemma 3.5, we control (51) as follows: with probability at least $1 - \delta$, it holds that

$$\left\| (x - y) - \tilde{\eta} \sum_{k=1}^K \left\langle \nabla \bar{\ell}(x) - \nabla \bar{\ell}(y), u_{t,k} \right\rangle u_{t,k} \right\|^2 \tag{141}$$

$$\overset{(a)}{\leq} \left( \left\| (x-y) - \sum_{k=1}^K \langle x - y, u_{t,k} \rangle u_{t,k} \right\|_2 + \left\| \sum_{k=1}^K \langle x - y, u_{t,k} \rangle u_{t,k} - \tilde{\eta} \sum_{k=1}^K \left\langle \nabla \bar{\ell}(x) - \nabla \bar{\ell}(y), u_{t,k} \right\rangle u_{t,k} \right\|_2 \right)^2$$

$$= \left( \left\| (x-y) - \sum_{k=1}^K \langle x - y, u_{t,k} \rangle u_{t,k} \right\|_2 + \left\| \sum_{k=1}^K \langle \phi(x) - \phi(y), u_{t,k} \rangle u_{t,k} \right\|_2 \right)^2 \tag{142}$$

$$\overset{(b)}{\leq} \left( \|x - y\|_2 \left( \frac{K}{2d+6} \log\left(\frac{4d}{\delta}\right) + \left( 0.15 - \frac{\log(2/\delta)}{K} \right)_+ \cdot \left( 1 - \frac{K}{20\left(d + 2 + \sqrt{d-1}\right)} \right)^2 \right)^{1/2} + \right.$$

$$\|\phi(x) - \phi(y)\|_2 \left( \frac{K}{2d+6} \log\left(\frac{4d}{\delta}\right) \right)^{1/2} \right)$$

$$\overset{(c)}{\leq} \|x - y\|_2 \cdot \left( \left( \frac{K}{2d+6} \log\left(\frac{4d}{\delta}\right) + \left( 1 - \left( 0.15 - \frac{\log(2/\delta)}{K} \right)_+ \cdot \frac{K}{20\left(d + 2 + \sqrt{d-1}\right)} \right)^2 \right)^{1/2} + \right.$$

$$c^2 \left( \frac{K}{2d+6} \log\left(\frac{4d}{\delta}\right) \right)^{1/2} \right)$$

where (a) follows from the triangular inequality, (b) holds due to Lemma K.4 the norm-sub-Gaussian property of $\langle \phi(x) - \phi(y), u_{t,k} \rangle u_{t,k}$, and (c) follows from the Lipschitzness of $\phi(\cdot)$. This yields a concentration bound on the generalized Lipschitzness of $\Psi_t$ as in Lemma 3.5, and hence can be plugged into Theorem 3.2 to obtain a Rényi DP bound.

**Discussion.** Compared to orthonormal update directions (as in Theorem 3.2), using i.i.d. update rules not only complicates the analysis significantly but also results in a substantially larger generalized Lipschitz constant $c_1$. In fact, even when the function $\phi$ has a small Lipschitz constant, the corresponding $c_1$ cannot be reduced below 1. This increase in $c_1$ stems from two main factors. First, the cross terms in the decomposition of (141) are no longer orthogonal and thus do not vanish, requiring the use of the triangle inequality to bound them. Second, the term $\sum_k \langle x - y, u_{t,k} \rangle u_{t,k}$ no longer admits a closed-form density, forcing the analysis to rely on general concentration inequalities, which can potentially be loose. While the bound we present may not yield the tightest possible constant, we believe that for i.i.d. updates, the Lipschitz constant of $\Psi_t$ (i.e., $c_1$ in Theorem 3.2) is fundamentally lower-bounded by 1.

