# OpenReview forum: "Privacy Amplification in Differentially Private Zeroth-Order Optimization with Hidden States"
_ICML.cc/2026/Conference — ICML 2026 regular_

### Official Review · Reviewer_uwA7 · 2026-03-11

**Soundness:** 2
**Presentation:** 3
**Significance:** 2
**Originality:** 3
**Overall Recommendation:** 4
**Confidence:** 4

**Summary:**

This paper studies hidden-state differential privacy (DP) for zeroth-order optimization. The motivation is DP fine-tuning, when first-order per-example gradients are too memory-expensive to compute. The paper proposes a hybrid-noise Noisy-ZOGD method and claims the first privacy-amplification-by-iteration style analysis for DP zeroth-order optimization.

**Compliance With Llm Reviewing Policy:**

Affirmed.

**Final Justification:**

The authors have addressed some of my concerns.

**Key Questions For Authors:**

See weaknesses.

**Limitations:**

yes

**Strengths And Weaknesses:**

Strengths:

The paper addresses an important and timely problem. DP zeroth-order methods are useful for memory-constrained fine-tuning, and extending hidden-state privacy analysis to zeroth-oder optimization is an interesting question. The hybrid noise design is also interesting, and the observation that orthonormal multi-direction updates can help privacy under hidden-state analysis is useful.

Weaknesses:

1. The proof of Theorem 3.2 seems to have some gaps. In particular, Definition 3.5 and Lemma 3.6 require a fixed ($c_1$, $c_2$)-generalized Lipschitz map. But in Lemma 3.7, the claimed $c_1$ depends on $\sum_k \upsilon_k$ and $\sum_k \gamma_k$, which in turn depend on $A = \frac{x-y}{||x-y||}$ and $B = \frac{\phi(x)-\phi(y)}{||\phi(x)-\phi(y)||}$. Lemma 3.8 then gives a high-probability bound only for fixed $A$, $B$. In the proof of Theorem 3.2, step (d) appears to use this as if there were an event which $\hat{\psi}$ is gloabally ($c_1$, $c_2$)-generalized Lipschitz, which does not seem to follow from the stated lemmas.

2. The practical relevance of the proposed paper is limited by the assumptions. The analysis relies on per-example smoothness and Lipschitzness, a bounded-domain projection, and in the closed-form corollary a strongly convex setting. The authors themselves acknowledge that the result is "not ready to be directly applied to LLM fine-tuning", which weakens the practical motivation.

3. The paper contains privacy-accounting figures, but there is no training experiments showing that the proposed hybrid noise, orthonormal directions, or larger $K$ actually improve privacy-utility or privacy-compute tradeoffs in practice.

4. The paper argues that a larger $K$ can help privacy under hidden-state analysis, but the closed-form result also requires $K$ to satisfy a lower bound that can be restrictive when conditioning is poor. It would be good to have a clearer discussion of the practically relevant regime of $K$, especially since larger $K$ directly increases the forward-pass cost.

5. The related work discussion is incomplete in a way that overstates novelty. The paper currently positions itself mainly against Zhang et al. (DPZero), Tang et al. (DP-ZO), Liu et al. (DP-ZOSO), and Gupta et al., and then claims that the privacy benefit of leveraging $K > 1$ zeroth-order gradients and choosing orthonormal directions was not studied in prior literature. It also states that prior DP zeroth-order studies avoided large $K$ because of the composition-based privacy cost. However, in Bao et al., "Unlocking the Power of Differentially Private Zeroth-order Optimization for Fine-tuning LLMs" (USENIX Security 2025), the privacy-utility benefit of using $K > 1$ has been investigated. So the novelty of the current submission should be re-positioned.

---

> ### Author Rebuttal · Authors · 2026-03-31
>
> We thank Reviewer uwA7 for their helpful comments. We address the misunderstanding in the raised weaknesses below.
>
> ## W1: Questions regarding the proof of Theorem 3.2.
> We sincerely thank reviewer uwA7 for pointing out this issue. Indeed, the pointwise bound implied by Lemma 3.8 is insufficient to establish the global generalized Lipschitz condition required for inequality (d). This is therefore a flaw in our original proof.
> Fortunately, the same DP guarantee can still be obtained with a careful modification of the theorem and its proof. The key idea is to avoid relying on a conditional RDP argument (on a high-probability event) followed by conversion to DP. Instead, we directly establish the DP guarantee by introducing a coupled auxiliary process $\widetilde{W}_t$ on the neighboring dataset $D'$. Recall that the original processes on $D$ and $D'$ are denoted by $W_t$ and $W_t^\prime$, respectively.
>
> Our approach proceeds in three steps. First, we bound the TV distance between $W_t$ and $\widetilde{W}_t$. This TV distance corresponds to the failure probability in the original Theorem 3.2, and contributes to the $\delta$ term in the final DP guarantee (denoted as $\delta_f$).
>
> Second, we bound the Rényi divergence between $\widetilde{W}_t$ and $W_t^\prime$ using standard Gaussian mechanism arguments.
>
> This yields the same RDP bound as in the original Theorem 3.2 (arising from the same constrained optimization), which we denote by $\rho_\alpha$.
>
> Finally, applying the standard RDP-to-DP conversion, we obtain that for any $\delta_p \in (0,1)$, the mechanism satisfies
> $$\left(\rho_\alpha + \frac{\log(1/\delta_p)}{\alpha - 1},\ \delta_p + \delta_f\right)\text{-DP}.$$
> This revised proof completely avoids the use of the pointwise bound (Lemma 3.8) in the shifted Rényi divergence analysis, and therefore no longer requires a global Lipschitz condition. Instead, we directly control the TV distance without relying on $W_\infty$ or the associated shifted analysis. As a result, the issue raised by reviewer uwA7 is fully resolved. ***Importantly, since the resulting DP bound remains unchanged, all other results and statements in the manuscript continue to hold.*** Since OpenReview does not render the LaTeX equations well, as well as having a limited character count, we include the revised theorem as well as its full proof in the link: https://anonymous.4open.science/r/Corrected_Proof_of_DPZOGD-B8B2/DPZOGD_proof_fix.pdf
>
> ## W2,3: Practical relevance of our work.
>
> We agree that there are still gaps to apply our hidden-state DP-ZOGD results to LLM training, which we have clearly stated as a limitation of this theoretical work. However, the same limitation exist even for prior works that studies hidden-state DP-GD (Altshular \& Talwar 2022, 2024; Ye \& Shokri, 2022; Chien \& Li, 2025 and more). We believe our theoretical contribution alone is sufficient to be published for the community.
>
> ## W4: Discussion about $K$.
>
> Thanks for the comment. We agree that larger $K$ can increase the computation, which is what we have discussed in page 5. As we demonstrated in Figure 1 (a), it is possible to use $K=200$ under $d=n=100000$. We believe this is still very efficient compared to the first-order method.
>
> ## W5: Comparison with Bao et al. (USENIX Security 2025)
>
> Thank you for bringing up this recent related work that we were previously not aware of. However, there are several key differences between our work and Bao et al. First, Bao et al. study DP-ZOGD under public state analysis, while our work focus on the hidden-state analysis. Second, the Bao et al. use ***independent*** Gaussian vectors as update directions, where we propose to use orthonormal basis (sampled from the Stiefel manifold) and we also show that i.i.d. directions lead to worse results in Appendix J. Third, we also study the hybrid noise mechanism, while Bao et al. focus on the isotropic noise only. Nevertheless, we agree to include Bao et al. in the related work section and mentioned that they also studied the benefit of $K\geq 1$, but from a very different angle (i.e., how increasing $K$ leads to a better ***per-iteration*** clipping error). It is an interesting future direction to see if the results of Bao et al. can be combined with our analysis.

---

> > ### Author Rebuttal · Reviewer_uwA7 · 2026-04-01
> >
> > I appreciate the authors’ candid response, especially on W1 and W5. My updated comments on W1-W4, based on the rebuttal text itself and not the external URL, are as follows.
> >
> > For W1, I appreciate that the authors explicitly acknowledge that the original proof of Theorem 3.2 has a flaw. However, I do not consider this point fully addressed from the rebuttal alone. The rebuttal gives only a high-level sketch of a different proof strategy, while the actual corrected theorem/proof is placed in an external file. Per the conference policy, reviewers are not expected to follow external URLs in the author response, so I cannot verify the fix. This matters because Theorem 3.2 is the paper’s central hidden-state privacy result, and Corollary 3.3 and the main privacy-saturation claims are built on top of it. I don't think I can simply assume that "all other results and statements continue to hold" without a self-contained correction in the rebuttal.
> >
> > For W2/W3, my concern is largely unchanged. The rebuttal mainly argues that the work is theoretical and that similar theory-practice gaps also appear in prior hidden-state DP papers. That is a reasonable defense of the paper’s scope, but it does not really address the specific concern I raised: the current assumptions are still fairly restrictive, and the paper still does not include end-to-end training experiments validating the claimed practical benefits of the hybrid noise, orthonormal directions, or larger $𝐾$. The paper itself also states that the theory is "not ready to be directly applied to LLM fine-tuning", which makes the practical motivation weaker than what the introduction suggests.
> >
> > For W4, I view the response as only partially addressing the issue. The example is useful, showing that $K=200$ is possible in one plotted regime, but my concern was more general: the paper would benefit from a clearer quantitative discussion of when the lower bound on $K$ in Corollary 3.3 is practically mild instead of restrictive, and how the privacy gain compares to the extra forward-pass cost as $K$ increases. That tradeoff still feels underexplained.

---

> > > ### Author Response · Authors · 2026-04-01
> > >
> > > We thank reviewer uwA7 for their follow-up question. Below, we include details of our fix.
> > >
> > > The construction of the auxiliary process is as follows. We start at $\widetilde W_\tau = W_\tau'$. Then let $Y_t, Z_t$ denote the scalar and isotropic noise for process $W_t$ as before. Next, we define the scalar-noise shift.
> > >
> > > $$b_t := \widehat \psi_{D,t}(\widetilde W_t) - \widehat \psi_{D',t}(\widetilde W_t).$$
> > >
> > > This scalar noise shift serves the same purpose as equation (7), where $b_t$ is the difference of execution on $D,D'$ with the same input argument. Next, we define the isotropic-noise shift.
> > >
> > > $$d_t := \widehat \psi_{D,t}(W_t) - \widehat\psi_{D,t}(\widetilde W_t).$$
> > >
> > > Then, we define the measurable vector
> > >
> > > $$v_t := \min(a_t,(||d_t||-z_{t+1})_+)\dfrac{d_t}{||d_t||},\;\text{for }d_t\neq 0,\;v_t=0\text{ if }d_t=0.$$
> > > Then $||v_t|| \leq a_t$ always. Together we define $\widetilde W_t$ as follows
> > >
> > > $$\widetilde W\_{t+1} := \Pi_{B_R}[\widehat\psi_{D',t}(\widetilde W_t) + Y_t + b_t + Z_t + v_t] = \Pi_{B_R}[\widehat \psi _{D,t}(\widetilde W_t) + Y_t  + Z_t + v_t]$$
> > >
> > > Note that $Y_t, Z_t$ here are coupled with those on $W_t$. Then we can define the success event $\mathcal{G}_t$ similar to before. That is, $\mathcal{G}_t$ is the event that the following holds.
> > >
> > > $$  || \widehat \psi_{D,t}(W_t) - \widehat\psi_{D,t}(\widetilde W_t) || \leq \bar{c}_1 ||W_t - \widetilde W_t|| + c_2$$
> > >
> > > where $\bar{c}_1  = \sqrt{1-(1-c^2)\frac Kd+\vartheta (1+c^2)\frac Kd}$ (i.e., the upper bound stated in Lemma 3.8.) By Lemma 3.8, we know that the condition on $\mathcal{F}_t$ (i.e., the sigma-field generated by all randomness up to time $t$ in the coupled construction), we have
> > >
> > > $$P(\mathcal G_t^c | \mathcal F_t) \leq p_\vartheta,$$
> > >
> > > where $p_\vartheta$ is the error probability stated in Lemma 3.8. Now define $\mathcal H_t$ be the event such that the following holds
> > >
> > > $$ \|W_t - \widetilde W_t\| \leq z_t.$$
> > >
> > > We already know that $H_{\tau}$ holds almost surely by Lemma 3.9. Then on event $\mathcal{H}_t \cap \mathcal{G}_t$, we have
> > >
> > > $ ||d_t|| \leq \bar{c}_1 z_t + c_2 \leq  z\_{t+1}+a_t,$
> > >
> > > where the last inequality is by the constraint of the optimization stated in the original Theorem 3.2. By construction of $v_t$, this implies
> > >
> > > $$\|d_t-v_t\|\leq z\_{t+1}.$$
> > >
> > > Then by the fact that projection is $1$-Lipschitz, we have
> > >
> > > $$||W\_{t+1} - \widetilde W\_{t+1}||\leq ||d_t - v_t||\leq z\_{t+1}.$$
> > >
> > > So $\mathcal{H}_t \cap \mathcal{G}_t \subseteq \mathcal{H}\_{t+1}$. Inductively,
> > >
> > > $$\cap\_{s=\tau}^t \mathcal{G}_s \subseteq \mathcal{H}\_{t+1},\;\text{for every }t\geq \tau.$$
> > >
> > > In particular, on $\mathcal{G} = \cap\_{s=\tau}^{T-1} \mathcal{G}_s$, we have $\mathcal{H}_T$, and since $z_T=0$, this implies
> > >
> > > $$W_T = \tilde{W}_T\;\text{on }\mathcal{G}.$$
> > >
> > > Moreover,
> > >
> > > $$P(\mathcal{G}^c)\leq \sum\_{t=\tau}^{T-1}P(\mathcal{G}_t^c) \leq (T-\tau)p\_\vartheta = \delta_f.$$
> > >
> > > Therefore, by the coupling inequality,
> > >
> > > $$TV(W_T,\tilde{W}_T)\leq P(W_T \neq \tilde{W}_T) \leq\delta_f. $$
> > >
> > > This completes the first half of the proof.
> > >
> > > For the RDP part, note that comparing $\widetilde W_t$ and $W_t'$, we have two shifts $b_t,v_t$. By their definition, we know that $||b_t||\leq \frac{2\eta\Delta}{\sqrt{K} n}$ and $||v_t|| \leq a_t$ always. These shifts will be taken care of by the directional noise part $Y_t$ and the isotropic noise part $Z_t$, respectively, via strong composition for Rényi divergence and the Gaussian mechanism. More specifically, for the scalar part, we have
> > >
> > > $$\frac{\alpha ||b_t||^2}{2(\eta^2\beta_t\sigma^2/K)}
> > >   \le
> > >   \frac{\alpha}{2\beta_t\sigma^2}\left(\frac{2\Delta}{n}\right)^2.$$
> > >
> > > For the isotropic part, we have
> > >
> > > $$\frac{\alpha ||v_t||^2}{2(\eta^2(1-\beta_t)\sigma^2/d)}
> > >   \le
> > >   \frac{\alpha d a_t^2}{2\eta^2(1-\beta_t)\sigma^2}$$
> > >
> > > The resulting RDP bound is $\rho_\alpha$. That is,
> > >
> > > $$D_\alpha(\widetilde W_T || W_T')\leq \rho\_\alpha.$$
> > >
> > > This completes the second half of the proof.
> > >
> > > Finally, standard RDP-to-DP conversion gives, for every fixed $\delta_p \in(0,1)$ and every measurable output set $S$,
> > >
> > > $$P(\widetilde W_T \in S)
> > >   \le
> > >   \exp\left(
> > >     \rho_\alpha+\frac{\log(1/\delta_{p})}{\alpha-1}
> > >   \right)P(W_T^\prime \in S)
> > >   +
> > >   \delta_{p}.$$
> > >
> > > Also, from the first part of the result:
> > >
> > > $$P(W_T \in S)\le P(\widetilde W_T \in S)+\delta_{f}$$
> > >
> > > Combining the inequalities,
> > >
> > > $$P(W_T \in S)
> > >   \le
> > >   \exp\left(
> > >     \rho_\alpha+\frac{\log(1/\delta_{p})}{\alpha-1}
> > >   \right)P(W_T^\prime \in S)
> > >   +
> > >   \delta_{p}+\delta_{f}.$$
> > >
> > > The same argument with $D$ and $D'$ exchanged gives the reverse DP inequality (due to replacement DP). Hence, the mechanism is
> > >
> > > $$\left(
> > >     \rho_\alpha+\frac{\log(1/\delta_{p})}{\alpha-1},
> > >     \delta_{p}+\delta_{f}
> > >   \right)\text{-DP}.$$
> > >
> > > ===
> > >
> > > We hope this provides enough detail regarding our fix. For the other comments, we feel they are more subjective, and we appreciate reviewer uwA7 for sharing their thought and respect them.

---

### Official Review · Reviewer_EBQj · 2026-03-12

**Soundness:** 4
**Presentation:** 3
**Significance:** 3
**Originality:** 3
**Overall Recommendation:** 5
**Confidence:** 3

**Summary:**

The paper considers the scenario where a ML model is finetuned with private data in a centralized non-distributed setting. The goal is to apply a DP mechanism so that the final model parameters do not (or little) leak information.

Rather than using DP-SGD which requires a costly per sample gradient clipping step, zeroth-order optimization is used where only forward steps are required. The paper employs (1) a random K dimensional update each step to reduce variance (that is K orthonormal perturbation directions are chosen at random), (2) Gaussian perturbation on the stepsize (this offers a DP guarantee that can be composed over iterations), and (3) a small isotropic Gaussian noise component to enable the shifted-reduction analysis (this allows application of the shift reduction lemma).

The main idea is to consider some iteration tau and notice that

(A) noise (3) allows a DP guarantee of the current model outputted at iteration tau. During next iterations this is amplified, i.e., we have “privacy amplification by iteration” because of the hidden optimization trajectory till the final iteration T. Here, the shift reduction lemma is used.

(B) During the last T minus tau iterations more private data is used and the noise (2) allows for a standard DP composition argument.
Combination of (A) and (B) allows one to balance these two DP contributions and minimize over tau. In particular, this makes the final bound independent of T (as one can always chose tau=T-constant).

The analysis shows that increasing K and using an orthonormal set of K perturbation directions amplifies privacy even more and leads to less leakage, i.e., eps = O(1/sqrt{K}). Increasing K makes the optimization more costly, hence, it allows a trade-off between complexity and privacy.

Contributions: The paper shows a new hybrid DP noise mechanism for zeroth-order optimization. This allows application of the shift reduction lemma and leads to an opportunity to balance/minimize the contributions to privacy loss by combining two separate analysis methods (composition and shift reduction).  The final result shows how the number of perturbation directions allows a trade-off between privacy loss and complexity.

**Compliance With Llm Reviewing Policy:**

Affirmed.

**Final Justification:**

I am fully satisfied with the author's answers and results in the paper.

**Key Questions For Authors:**

In order to reach a similar utility compared to a plain SGD approach, how many times more iterations (times K) does zeroth-order optimization require? Does this generally push one to the ceiling of the curves in Figure 1?

How low can eps be pushed in a practical setting? For what type of learning task does eps=0.5 lead to a parameter setting with reasonable complexity and utility? Is eps=0.05 possible? (If this is not yet clear, then the paper may mention this is well.)

**Limitations:**

yes

**Strengths And Weaknesses:**

The paper is technically sound – all the intuition matches and the reasoning in the main body is proper. I did not check in detail the proofs in the appendix. The main body does show how the several puzzle pieces fits neatly.

The paper reads very well. Theorem 3.2 mentions P(G) for the first time and you will want to spend a couple of words explaining this probability.

The paper addresses how zeroth-order optimization can be combined with “privacy amplification by iteration” leading to an improved bound on privacy leakage. The paper introduces a new hybrid noising mechanism.

For future work, it remains an open problem whether the isotropic noise (3) is information theoretically necessary. The appendix shows an extension to minibatches.

The weakness of the paper is that it is restricted to the centralized setting where a single authority uses its own collected private data for finetuning a ML model. In this sense this work is not applicable in a FL setting.

Another weakness of the paper is that, even though the theoretical bounds are nicely plotted, an actual simulation to understand how the added noises (2) and (3) affect utility and how parameter K affects complexity in practice is missing. The plots represent the expectation of most other papers in literature, i.e., a reported eps=0.25 for K=2000 and eps=0.5 for K=500 in Figure 1. It is important to realize that eps=0.5 actually proves very little privacy (the reduced hypothesis testing problem for membership inference has a significant advantage over random guessing). Like most ML literature this is not talked about. Nevertheless the theoretical advantage provided by the paper is important (and for future work, is it possible to design added “tricks” to close the privacy-utility-complexity gap even more – the paper may want to call out that this is a step in the right direction but more needs to be done).

---

> ### Author Rebuttal · Authors · 2026-03-31
>
> We thank Reviewer EBQj for their positive comments and thoughtful feedback. Below, we address the raised questions.
>
> ## Q1: In order to reach a similar utility compared to a plain SGD approach, how many times more iterations (times K) does zeroth-order optimization require? Does this generally push one to the ceiling of the curves in Figure 1?
>
> This is a great question! Roughly speaking, for ZOGD using $K$ orthonormal directions, we need $O(d/K)$ more iterations compared to GD in the worst case (for smooth losses). As discussed in the DPZero for case $K=1$, this overhead seems inevitable for the worst-case. So, as Reviewer EBQj correctly mentioned, this generally pushes one to the ceiling of the curves in Figure 1. Note that whenever we reach the flat region for each line in Figure 1, our results provide a privacy bound that is strictly better than both standard composition as well as output perturbation. This also highlights the reason why a hidden-state DP analysis type of result is even more important in the DP-ZOGD scenario. At last, we want to emphasize that while there is a worst-case showing that ZOGD in general needs $O(d)$ more iterations, in practice, we may not encounter such a worst-case. So it is possible that DP-ZO(S)GD can be more favorable in terms of privacy-utility trade-off compared to DP-(S)GD, as demonstrated in the LLM finetuning setting in DPZero and Tang et al., 2024.
>
> ## Q2 How low can eps be pushed in a practical setting? For what type of learning task does eps=0.5 lead to a parameter setting with reasonable complexity and utility? Is eps=0.05 possible? (If this is not yet clear, then the paper may mention this is well.)
>
> For smooth loss, we do provide the corresponding utility bound similar to those of DPZero. See Appendix I, equation (127). For a practical setting, this is not yet clear. We have mentioned in the future direction section that our results are not yet ready to be applied to DP LLM training. We will further emphasize it in our revision. Thank you for the question.

---

> > ### Author Rebuttal · Reviewer_EBQj · 2026-04-03
> >
> > Thank you for your (satisfactory) answers.
> >
> > I recently saw the paper "Unlocking the Power of Differentially Private Zeroth-order Optimization for Fine-tuning LLMs," by
> > Ergute Bao∗, Yangfan Jiang†, Fei Wei∗, Xiaokui Xiao†, Zitao Li∗, Yaliang Li∗, Bolin Ding at USENIX 2025.
> >
> > I would love to hear your thought about how your work relates to this and whether this paper is also worth mentioning in related work?

---

> > > ### Author Response · Authors · 2026-04-03
> > >
> > > Thank you for bringing up this recent related work that we were previously not aware of. There are several key differences between our work and that of Bao et al. First, Bao et al. studied DP-ZOGD under public state analysis, while our work focuses on the hidden-state analysis. Second, the Bao et al. use independent Gaussian vectors as update directions, where we propose to use orthonormal basis (sampled from the Stiefel manifold), and we also show that i.i.d. directions lead to worse results in Appendix J. Third, we also study the hybrid noise mechanism, while Bao et al. focus on the isotropic noise only.
> > >
> > > Nevertheless, we agree that we should include Bao et al. in the related work section and mention that they also studied the benefit of $K>1$, but from a very different angle (i.e., how increasing $K$ leads to a better ***per-iteration*** clipping error). It is an interesting future direction to see if the results of Bao et al. can be combined with our analysis.

---

### Official Review · Reviewer_myef · 2026-03-12

**Soundness:** 4
**Presentation:** 3
**Significance:** 3
**Originality:** 3
**Overall Recommendation:** 5
**Confidence:** 4

**Summary:**

This paper studies the privacy guarantees of zeroth-order methods for DP training in the hidden state threat model. This threat model is related to Privacy Amplification by Iteration (PABI), where the parameters are not revealed to the attacker during training. The PABI literature is prolific  for first-order training. A natural research question is whether PABI persists for zeroth-order methods.

The paper gives a positive answer for a modified mechanism adapted from existing DP zeroth-order methods. The proposed mechanism, Noisy-ZOGD, combines isotropic noise and univariate noise along random directions. The authors prove the convergence of the privacy loss of Noisy-ZOGD under a strong convexity assumption. They also show the surprising result that privacy loss decays with the update dimension $K$. The paper also discusses the nonconvex setting and proposes a numerical accounting method tailored to the nonconvex case.

**Compliance With Llm Reviewing Policy:**

Affirmed.

**Final Justification:**

After the rebuttal, my concerns have been addressed. Therefore, I decide to keep my score.

**Key Questions For Authors:**

- Can the authors discuss the PABI bounds obtained under smoothness assumptions only? How do these bounds compare to the worst-case adversaries of [1,2]?
- It is unclear to me whether the hybrid mechanism is required to obtain amplification results. I have understood that isotropic noise is necessary to leverage the shifted divergence framework. Can the authors comment on the possibility of PABI bounds in the case $\beta_t = 0$, through other techniques?

**Limitations:**

yes

**Strengths And Weaknesses:**

## Strengths

- This paper fills a gap in the PABI literature, which has exclusively focused on first-order methods.
- The technical development appears to be non-trivial, and some developed tools, such as Lemma $3.6$ for the shifted divergence framework, might be of independent interest.
- The paper discusses the flaws of existing mechanisms and proposes a mechanism which enjoys privacy loss convergence.

## Weaknesses

- My main concern relates to the PABI bounds obtained under nonconvexity assumptions. It has been shown in the literature that one cannot hope to obtain general PABI bounds in nonconvex scenarios [1,2].
- The paper does not prove amplification for existing zeroth-order DP mechanisms. Instead, it proves amplification for a modified hybrid mechanism.

## References

[1] M. S. M. S. Annamalai. "It’s Our Loss: No Privacy Amplification for Hidden State DP-SGD With Non-Convex Loss." (2024). In AISec.

[2] T. Cebere, A. Bellet, and N. Papernot. "Tighter Privacy Auditing of DP-SGD in the Hidden State Threat Model." (2024). In International Conference on Learning Representations.

---

> ### Author Rebuttal · Authors · 2026-03-31
>
> We thank Reviewer myef for their positive comments and thoughtful feedback. Below, we address the raised questions as well as provide clarifications for weaknesses.
>
>
> ## Q1,W1: My main concern relates to the PABI bounds obtained under nonconvexity assumptions. It has been shown in the literature that one cannot hope to obtain general PABI bounds in nonconvex scenarios [1,2].
>
> This is a great comment! Note that our results (or any existing PABI results without the convexity assumption, like those presented in Chien and Li, 2025) do not contradict the results of privacy auditing for the last iterations, like [1,2] (which we have cited both). Currently, to the best of our knowledge, any PABI results still require some structural assumptions on the loss. For example, in this work, we need smoothness, and in Chien and Li, 2025, they at least need H\”older continuous gradient. None of these structural assumptions holds for the case studied in [1,2], where they focus on auditing neural networks. In fact, we do believe that it is impossible to obtain PABI-type bounds for general loss (i.e., neural nets studied in [1,2]). Yet, it also gives us a hint that maybe it is worth investigating more on “neural nets with nice structural properties” such as smoothness due to privacy concerns. For instance, Lipschitz networks are studied in the literature, albeit they mainly focus on a robustness perspective (to input) instead of to the weights. At last, we also provide the numerical results for solving the optimization problem for the privacy bound stated in Theorem 3.2 in Figure 2 in the Appendix. It shows that indeed our Theorem 3.2 provides a nontrivial hidden-state DP bound compared to output perturbation and standard composition.
>
> ## Q2: It is unclear to me whether the hybrid mechanism is required to obtain amplification results. I have understood that isotropic noise is necessary to leverage the shifted divergence framework. Can the authors comment on the possibility of PABI bounds in the case $\beta_t=0$, through other techniques?
>
> This is another great question. Indeed, in theory, one may still derive a PABI type of result for the case $\beta_t=0$ (i.e., using only isotropic noise). However, as shown in DPZero as well as our utility analysis in Appendix I, using isotropic noise will “hurt” utility more severely compared to directional Gaussian noise. Indeed, consider the first iteration only with $K=1$, the only difference between the two adjacent training processes $w_t,w_t^\prime$ only appears in the direction $u$. So we should use a directional Gaussian noise to protect such a difference. If one uses isotropic Gaussian noise, the noise added in the subspace $u^\perp$ only degrades utility but provides no benefit in terms of privacy. In fact, this is why we present the public-state result (Theorem 3.1) first to argue why we want to consider directional Gaussian noise. If one really uses pure isotropic Gaussian noise, the corresponding bound in equation (9) will be worse, similar to the results presented in Theorem 3.1. This is the reason why we adopted the hybrid noise strategy instead. We only want to use the least possible amount of isotropic Gaussian noise for the shift reduction lemma (Lemma 3.4), and the rest of the part can be taken care of by the directional Gaussian noise.
>
>
> ## W2: The paper does not prove amplification for existing zeroth-order DP mechanisms. Instead, it proves amplification for a modified hybrid mechanism.
>
> Fair point. As we discussed in our future directions section, currently, how to provide PABI-type results without isotropic noise is an open question. While we conjecture that establishing a version of Lemma 3.6 without isotropic noise is impossible, we do not claim that isotropic noise is information-theoretically necessary. Further study regarding this is an interesting direction.

---

> > ### Author Rebuttal · Reviewer_myef · 2026-04-03
> >
> > Thank you for your answers. My questions have been fully addressed, and I think that this work is theoretically interesting.

---

### Official Review · Reviewer_HPJ3 · 2026-03-12

**Soundness:** 3
**Presentation:** 4
**Significance:** 3
**Originality:** 3
**Overall Recommendation:** 5
**Confidence:** 4

**Summary:**

This paper studies zeroth-order gradient descent (ZOGD) in the differentially private setting. Existing DP-ZO optimization algorithms mainly focus on investigating the iteration-wise privacy guarantee, leaving last-iteration privacy guarantee unexplored. This paper focus on providing a theroretical analysis on the privacy guarantee for last-iteration noisy-ZOGD algorithm (or hidden-state privacy anlysis).

The paper first provide a generalized update formulation for noisy ZOGD for two different types of noise injection approach. Then it uses Renyi DP and Wasserstein distance to track the distribution shift during noisy ZOGD update. The theoretical results shows that noisy-ZOGD algorithm enjoys a similar bound compared to first-order DPGD algorithm in last-iterate DP.

**Compliance With Llm Reviewing Policy:**

Affirmed.

**Final Justification:**

Based on my initial comments and the rebuttals, I will keep my current score.

**Key Questions For Authors:**

1. Please provide a comparision between the last-iterate privacy guarantee between DP-ZOGD and DP-GD.

2. Is there an optimal choice of $K$ that can be derived from the theoretical result?

**Strengths And Weaknesses:**

Strength:
1. Clarity: the paper is clearly written. Sufficient details are provided to establish the privacy guarantee for the algorithm.
2. Novelty: The paper provides a novel privacy guarantee for last-iterate ZOGD algorithm. The proof techniqe follows existing track in proving last-iterate privacy guarantee but incooperate the dynamics of zeroth-order optimization in the distribution shift. The thoeretical results also reflects a novel observation on the benefit of using multiple directions in DO-ZOGD, which has not been studied in the previous paper.
3. Soundness: The paper provide detailed theoretical proofs in the appendix for the algorithm.

Weakness:
1. The paper require the per-sample gradient to be bounded by $\\Delta$ to avoid considering the impact of gradient clipping. It is interesting to see how clipping affects the overall privacy guarantee.

2. The paper considers vector $\{u\}$ to be drawn from Stiefel manifold. In practice, Gaussian distribution + orthorgonalization is also comminly used. The theorem did not consider how other distribution of the direction in ZOGD affects the output distribution.

3. ZOGD to ZOSGD. In practice, due to the large dataset size, tis it typically suggested to use stochastic/minibatch (zeroth-order) gradient instead of full gradient. It is unclear how would this change the result.

---

> ### Author Rebuttal · Authors · 2026-03-30
>
> We thank Reviewer HPJ3 for their positive comments and thoughtful feedback. Below, we address the raised questions as well as provide clarifications for weaknesses.
>
> ## Q1: Please provide a comparison between the last-iterate privacy guarantee between DP-ZOGD and DP-GD.
>
> Thanks for the great comment. First of all, we want to emphasize that our result (i.e., Corollary 3.3 for smooth strongly convex loss) holds for the update that uses both isotropic Gaussian noise as well as 1-dim Gaussian noise on the update directions $u_k$. Thus, the algorithm itself is very different from the DP-GD case, as studied by Altschuler and Talwar, 2022. Nevertheless, we provide the bounds below the comparison.
>
> DP-ZOGD:
> $$\varepsilon = O\left( \sqrt{\frac{\Delta^2 \log(1/\delta)}{n^2\sigma^2} \min(T,\frac{MRn\sqrt{d}}{K\Delta}}) \right)$$
>
> DP-GD (Remark 1.4, Altschuler and Talwar, 2022):
>
> $$\varepsilon = O(\sqrt{\frac{\Delta^2}{n^2\sigma^2}\log(1/\delta)\min(T,\frac{RM n}{\Delta })})$$
>
> There are several remarks for this comparison. First, for the DP-GD result, we set the step size to be $1/M$, where $M$ is the smoothness of the loss. In contrast, for DP-ZOGD we choose the step size to be $\eta = K/M$ (and the corresponding noise scaling may differ, as we do not calibrate the utility bound for DP-GD result). Second, our results require strong convexity, while the presented result for DP-GD only requires convexity. Third, we need the perturbation scale $\xi$ to be sufficiently small. Fourth, the presented DP-GD results require $\varepsilon \lessim 1$ according to Remark 1.4 of Altschuler and Talwar, 2022. Note that their results are mainly presented in terms of RDP.
>
> ## Q2: Is there an optimal choice of $K$ that can be derived from the theoretical result?
>
> As we discussed in the paragraph following Corollary 3.3, leveraging a larger $K$ is always beneficial in terms of privacy-utility tradeoff. However, it also introduces additional computation overhead compared to the case $K=1$ (i.e., DPZero). Thus, $K$ controls the computation-privacy/utility tradeoff. The optimal choice of $K$ depends on how many computations one is willing to pay per iteration.
>
> ## W1:  The paper require the per-sample gradient to be bounded by $\Delta$  to avoid considering the impact of gradient clipping. It is interesting to see how clipping affects the overall privacy guarantee.
>
> This is a great comment. While it may be hard to interpret the effect of $\Delta$ in our most general result, Theorem 3.2, we also provide the closed-form solution for $\varepsilon$ as a function of $\Delta$ for smooth and strongly convex losses. The effect of $\Delta$ can be clearly interpreted in Corollary 3.3, which has a similar effect as in the DP-GD case.
>
> ## W2: The paper considers vector $u$ to be drawn from Stiefel manifold. In practice, Gaussian distribution + orthorgonalization is also comminly used. The theorem did not consider how other distribution of the direction in ZOGD affects the output distribution.
>
> There might be a misunderstanding with regard to this comment. As we discussed on page 7, the second paragraph under Further Remarks, we also provide the analysis regarding the case of $u$ being drawn i.i.d. from Gaussian distribution (without orthogonalization), and the complete analysis is provided in Appendix J. We show that using orthonormal directions enables the application of tight quantile bounds from the beta distribution, which are critical to our privacy analysis. Nevertheless, we agree that exploring direction distributions other than Gaussian is an interesting future direction.
>
> ## W3: ZOGD to ZOSGD. In practice, due to the large dataset size, tis it typically suggested to use stochastic/minibatch (zeroth-order) gradient instead of full gradient. It is unclear how would this change the result.
>
> This is also a great comment, which is exactly the reason why we provide the mini-batch extension of our main results in Appendix H. As we discussed in the first paragraph under Further remarks, we provide the result of subsampling without replacement (i.i.d. across different iterations, the same setting as in Altschuler and Talwar, 2022 and Chien and Li, 2025) for ZOSGD, which is a generalization of Theorem 3.2. We agree that it is still an open problem for the other mini-batch strategy, such as the shuffled cyclic minibatch studied by Ye and Shokri, 2022 and Chien and Li, 2025. We left it as an interesting future work.

---

> > ### Author Rebuttal · Reviewer_HPJ3 · 2026-04-02
> >
> > My most questions have been addressed.
> >
> > A follow-up question: Combining the response for Q1 and Q2, it seems like by selecting $K = \\sqrt{d}$, the convergence rate of DP-ZOGD matches DP-GD (upto some constant factor). Is there still benefit of choosing larger $K$ beyond this point, indicating that DP-ZOGD can outperforms DP-GD?

---

> > > ### Author Response · Authors · 2026-04-03
> > >
> > > Thanks for the insightful question. The short answer is no, DP-ZOGD cannot outperform DP-GD. As we mentioned in our response, "the corresponding noise scaling may differ, as we do not calibrate the utility bound for DP-GD result". Indeed, intuitively, when $K=d$ under the orthonormal directions $u_{t,k}$, the "gradient" in DP-ZOGD "becomes" (in the limit $\xi \rightarrow 0$) the actual first-order gradient in DP-GD (i.e., it is just a change of basis, from $e_k$ to $u\_{t,k}$). So it should be impossible to outperform DP-GD.
> > >
> > > The bounds presented in the paper are mainly for fair comparison for $\beta_t \in [0,1]$ under DP-ZOGD context, but not DP-GD. Indeed, our DP-ZOGD update is
> > >
> > > $$w\_{t+1} = \Pi\_{\mathcal{B}\_R}[w_t - \frac{\eta}{K}\sum_{k=1}^K\hat{g}\_{t}(w_t;u_{t,k}) + \frac{\eta}{\sqrt{K}} \sum_{k=1}^K G\_{t,k}^{(1)} u\_{t,k} + \frac{\eta}{\sqrt{d}} G\_t^{(2)}]$$
> > >
> > > where $G_{t,k}^{(1)} \sim \mathcal{N}(0, \beta_t\sigma^2)$, and $G_t^{(2)} \sim \mathcal{N}(0, (1-\beta_t) \sigma^2 I_d)$.
> > >
> > > In contrast, the update stated in Altschuler and Talwar, 2022 is
> > >
> > > $$w\_{t+1} = \Pi\_{\mathcal{B}\_R}[w_t - \eta g_t + \eta G\_t]$$
> > >
> > > where $G_t \sim N(0,\sigma^2 I_d)$.
> > >
> > > So apparently, the scaling of the noise in these two updates is different (i.e., by comparing our special case $\beta_t = 0$ ot the first order update above), so that the corresponding $\sigma^2$ to achieve the same utility bound will also be different. While we do not provide a rigorous calibration with respect to the utility bound of DP-GD in the hidden-state context, if we view $\frac{1}{K}\sum_{i=1}^K \sum_{k=1}^K\hat{g}\_{t}(w_t;u_{t,k})$ being roughly the same effect as $g_t$, then we can see that the difference in noise factor is $\frac{1}{\sqrt{d}}$ (see, for example the case $K=d$), which implies that our bound at best recovers the privacy bound of the first-order result of Altschuler and Talwar, 2022 (up to some constant). That is one should replace $\sigma^2$ with $\frac{\sigma^2}{\sqrt{d}}$ in our DP-ZOGD bound when comparing with the bound of DP-GD given by Altschuler and Talwar, 2022.
> > >
> > > Of course, we have not rigorously proved this, as we do not calibrate DP-ZOGD and DP-GD with the actual utility bound in the paper. Also, a caveat is that our results require strong convexity, while the DP-GD bound of Altschuler and Talwar, 2022 in our first response only requires convexity. We will include this discussion in our revision. Thank you again for raising this insightful question.

---

### Decision · Program_Chairs · 2026-04-30

**Decision:**

Accept (regular)

**Comment:**

This paper studies hidden-state differential privacy for zeroth-order optimization and introduces a hybrid noise mechanism with orthonormal update directions. The main contribution is a theoretical analysis of privacy amplification-by-iteration for DP zeroth-order methods.

Overall, all reviewers ended up in favor of acceptance after the rebuttal period, with two initial weak points (proof concerns and practical limitations) being either corrected or satisfactorily addressed. One reviewer identified a flaw in the original proof of Theorem 3.2, which the authors acknowledged and fixed with a revised argument. The other remaining concerns were judged acceptable for a primarily theoretical contribution.

Therefore, I recommend to accept the paper, and I ask the authors to incorporate all promised updates to the final version.